# Regulation of Polyhomeotic Condensates by Intrinsically Disordered Sequences That Affect Chromatin Binding

**Ibani Kapur** [1,2], **Elodie L. Boulier** [1] **and Nicole J. Francis** [1,2,3,*]

1 Institut de Recherches Cliniques de Montréal, 110 Avenue des Pins Ouest, Montréal, QC H2W 1R7, Canada
2 Division of Experimental Medicine, McGill University, 1001 Decarie Boulevard, Montreal, QC H4A 3J1, Canada
3 Département de Biochimie et Médecine Moléculaire, Université de Montréal, 2900 Boulevard Edouard-Montpetit, Montréal, QC H3T 1J4, Canada
* Correspondence: nicole.francis@ircm.qc.ca

**Abstract:** The Polycomb group (PcG) complex PRC1 localizes in the nucleus in condensed structures called Polycomb bodies. The PRC1 subunit Polyhomeotic (Ph) contains an oligomerizing sterile alpha motif (SAM) that is implicated in both PcG body formation and chromatin organization in *Drosophila* and mammalian cells. A truncated version of Ph containing the SAM (mini-Ph) forms phase-separated condensates with DNA or chromatin in vitro, suggesting that PcG bodies may form through SAM-driven phase separation. In cells, Ph forms multiple small condensates, while mini-Ph typically forms a single large nuclear condensate. We therefore hypothesized that sequences outside of mini-Ph, which are predicted to be intrinsically disordered, are required for proper condensate formation. We identified three distinct low-complexity regions in Ph based on sequence composition. We systematically tested the role of each of these sequences in Ph condensates using live imaging of transfected *Drosophila* S2 cells. Each sequence uniquely affected Ph SAM-dependent condensate size, number, and morphology, but the most dramatic effects occurred when the central, glutamine-rich intrinsically disordered region (IDR) was removed, which resulted in large Ph condensates. Like mini-Ph condensates, condensates lacking the glutamine-rich IDR excluded chromatin. Chromatin fractionation experiments indicated that the removal of the glutamine-rich IDR reduced chromatin binding and that the removal of either of the other IDRs increased chromatin binding. Our data suggest that all three IDRs, and functional interactions among them, regulate Ph condensate size and number. Our results can be explained by a model in which tight chromatin binding by Ph IDRs antagonizes Ph SAM-driven phase separation. Our observations highlight the complexity of regulation of biological condensates housed in single proteins.

**Keywords:** Polycomb; condensate; phase separation; chromatin; *Drosophila*; intrinsically disordered region; sterile alpha motif

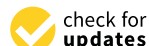



## 1. Introduction

The role of biomolecular condensates in the regulation of cellular processes, including gene expression, is increasingly appreciated. Phase separation, the demixing of molecules in solution to dense and dilute phases, may underlie many condensates that form in cells, providing a means to concentrate and segregate biomolecules [1–3]. However, while the core physical principle of phase separation may explain condensates, phase separation must be tightly regulated in time and space to generate condensates with biological functions. The material properties of condensates, which can include more or less viscous liquids and gel-like or even solid states, are also essential for their function [4,5]. Finally, biological condensates are not simple polymers in solvent systems; rather, they are highly complex structures existing in a complex solvent (i.e., the cytoplasm or nucleus) [6].

Understanding how complex condensates form in cells, identifying key proteins that drive their formation, and determining how the sequences of these proteins contribute to

condensates are essential. Many examples, as well as theory, have demonstrated how both structured domains and intrinsically disordered sequences can undergo phase separation under biologically relevant conditions [7]. Because weak, multivalent interactions are key to creating liquid condensates, intrinsically disordered regions (IDRs) are especially well-suited to undergo phase separation [1,7,8]. However, IDRs may lack the specificity needed to form relevant condensates in a complex cellular environment [9]. There are now several examples of proteins that drive condensate formation through a combination of structured domains and IDRs [10,11].

The Polycomb group (PcG) proteins are epigenetic regulators of gene expression that function by modifying chromatin structure [12–14]. Two main PcG multiprotein complexes, PRC1 and PRC2, are conserved across evolution [15]. These complexes modify histones (H2A119Ub and H3K27me3, respectively). PcG proteins also organize chromatin at a large scale, forming large, compacted chromatin domains [13,16]. This activity is distinct from histone modifications. PcG proteins themselves form visible condensates, and these condensates co-localize with PcG-regulated genes [13,16]. This suggests that the formation of PcG condensates and compacted chromatin domains are linked processes. PRC1 components, particularly Polyhomeotic (Ph) and Polycomb (Pc) (PHC and CBX in mammals), are implicated in condensate formation and chromatin organization [17,18].

Ph has a C-terminal sterile alpha motif (SAM) that is essential for its function and implicated in both condensate formation and chromatin organization. Ph SAM can form head-to-tail polymers [19], and this polymerization activity is important for Ph function [20]. Dominant negative Ph SAM mutations that disrupt the polymerization of wild-type Ph SAM also disrupt PcG clusters, lead to the loss of chromatin compaction, and disrupt gene regulation in both *Drosophila* and mammalian cultured cells and in developing mice [21,22]. The loss of Ph leads to the decompaction of chromatin in *Drosophila* cells, *Drosophila* embryos, and mouse embryonic stem cells [17,23,24]. Although SAM polymerization is its hallmark activity, genetic experiments indicate that Ph SAM has critical polymerization-independent functions because a *ph* transgene with a polymerization interface mutated can rescue some *ph* functions in *Drosophila* embryos but *ph∆SAM* cannot [25].

We recently showed that Ph SAM, in the context of a truncated version of Ph containing its three conserved domains called mini-Ph, undergoes phase separation with DNA or chromatin in vitro. Phase separation requires the SAM but does not strictly require polymerization [26]. Human PHC1 was also shown to undergo SAM-dependent phase separation in cells when induced to cluster through optogenetic manipulation [27]. Thus, phase separation could be the critical second Ph SAM activity. However, in cells, Ph forms many condensates while mini-Ph forms a single condensate; both types of condensates require the SAM [26]. Thus, we hypothesized that sequences N-terminal to mini-Ph regulate condensate formation through the SAM. Indeed, *Drosophila* Ph has ~1200 additional amino acids N-terminal to mini-Ph that have no recognizable domains and are not characterized. Most mammalian PHC isoforms (all of which have the mini-Ph region) also contain an uncharacterized sequence N-terminal to the SAM.

Here, we analyzed the N-terminal sequence of Ph, which is predicted to be intrinsically disordered. We divided the N-terminal sequence into three distinct IDRs based on sequence composition and complexity, and tested their effects on condensate formation in cells. We found that all three IDRs influenced condensate formation. The most prominent effects were mediated by a central, glutamine-rich IDR; the deletion of this region resulted in large, round condensates that excluded chromatin, similar to those formed by mini-Ph. The deletion of this IDR also strongly reduced chromatin association, suggesting that the effects of this sequence on condensate properties may be determined by chromatin binding. The other IDRs also affected condensate properties and chromatin binding, including an IDR that was previously shown to regulate Ph SAM aggregation through its O-linked glycosylation. We found that this IDR also negatively regulates the activity of the other two IDRs. Our results point to chromatin binding as

an important constraint on phase separation in cells and suggest that balance between the two has evolved to tune biological condensates.

## 2. Results

### 2.1. The N-Terminal Region of Drosophila Ph Is Predicted to Be Disordered and Comprises Three Distinct Regions

*Drosophila* melanogaster contains two tandem *ph* genes (*ph-p* and *ph-d*) that are highly similar and largely functionally redundant [28,29]. We analyzed Ph-p, referred to hereafter as Ph. Ph is a large protein (1589 amino acids), with three small conserved domains in its C-terminal region, namely, the HD1, FCS and SAM (Figure 1A). Previous work noted biased sequence composition in the region N-terminal to the conserved domains [30] and characterized an unstructured S-T-rich region that is both phosphorylated and heavily glycosylated by the O-GlcNac transferase (OGT). OGT is encoded by the PcG gene *super sex combs* (*sxc*) [25,31]. We used MetapredictV2 [32,33], which integrates disorder prediction and AlphaFold2 structure prediction, to analyze the sequence of Ph. This algorithm classified the entire N-terminal region of Ph upstream of the SAM as disordered (Figure 1A). This includes the FCS and HD1 domains, both of which have solved structures from human Ph homologues (PHC2 and PHC1, respectively) [34,35]. Although classified as disordered, these regions show a dip in disorder and a corresponding increase in structure. The structure of the HD1 was obtained in the presence of a (fused) binding partner; it therefore seems possible that this is an induced fold, and thus that the sequence is disordered in an isolated polypeptide. We used MetapredictV2 to analyze all three PHC sequences; in all cases, the FCS, but not the HD1, is predicted to be ordered (Figure S1), indicating that differences in the FCS sequence explain the difference in the predicted structures. A small possibly structured region is predicted around aa400 of Ph, and a larger region that corresponded to a Q-rich stretch (grey bars, Figure 1A) is predicted to form a long helix by AlphaFold2, as well as to form coiled coils (not shown). As recently discussed, predictions of coiled coil and helical structures from poly-Q regions are not well-supported by structural data [36], although the importance of the helix-forming ability of a PolyQ sequence in regulating phase separation has also been demonstrated [37]. Like any prediction, the disordered nature of the N-terminal sequences in Ph will need to be experimentally confirmed; we refer to the entire region as disordered, with the above caveats noted (Figure 1A).

Intrinsically disordered protein sequences often also contain low-complexity and compositionally biased sequences. To identify such regions in Ph and potential subregions in the large disordered region, we used the SEG and CAST [38] algorithms using the PlaToLoCo (Platform of Tools for Low Complexity) interface [39]. Although low-complexity sequence is present throughout the N-terminal region, the masked (i.e., repetitive) amino acids are different in different regions of the sequence. This allowed us to demarcate Ph1 (S-masked), Ph2 (Q-masked), and Ph3 (S+T masked) (Figure 1B). Ph1 did not show a clear enrichment for a single amino acid. Although serines are masked using the CAST algorithm, this indicates that they are repetitive, not over-represented. Q and H residues are slightly over-represented in Ph1. Ph2 is enriched in glutamine (36%), and Ph3 was enriched in threonine/serine (20% each) (Figure 1C,D). PLAAC was used to scan for prion-like amino acid composition in the protein sequence [40]. Consistent with its high glutamine content, Ph2 has high scoring predicted prion-like regions; both Ph1 and Ph3 have smaller predicted prion-like regions (Figure 1E). In summary, we refer to the three Ph regions as Ph(IDR)1-3. We used these boundaries to design constructs to test their functions. The full sequence of each IDR is provided in Figure S2. Although the sequence composition rationalized the boundaries chosen, they must be regarded as somewhat arbitrary. The boundaries split the disordered region into three roughly equal sized pieces, making them a good starting point for structure function analysis.

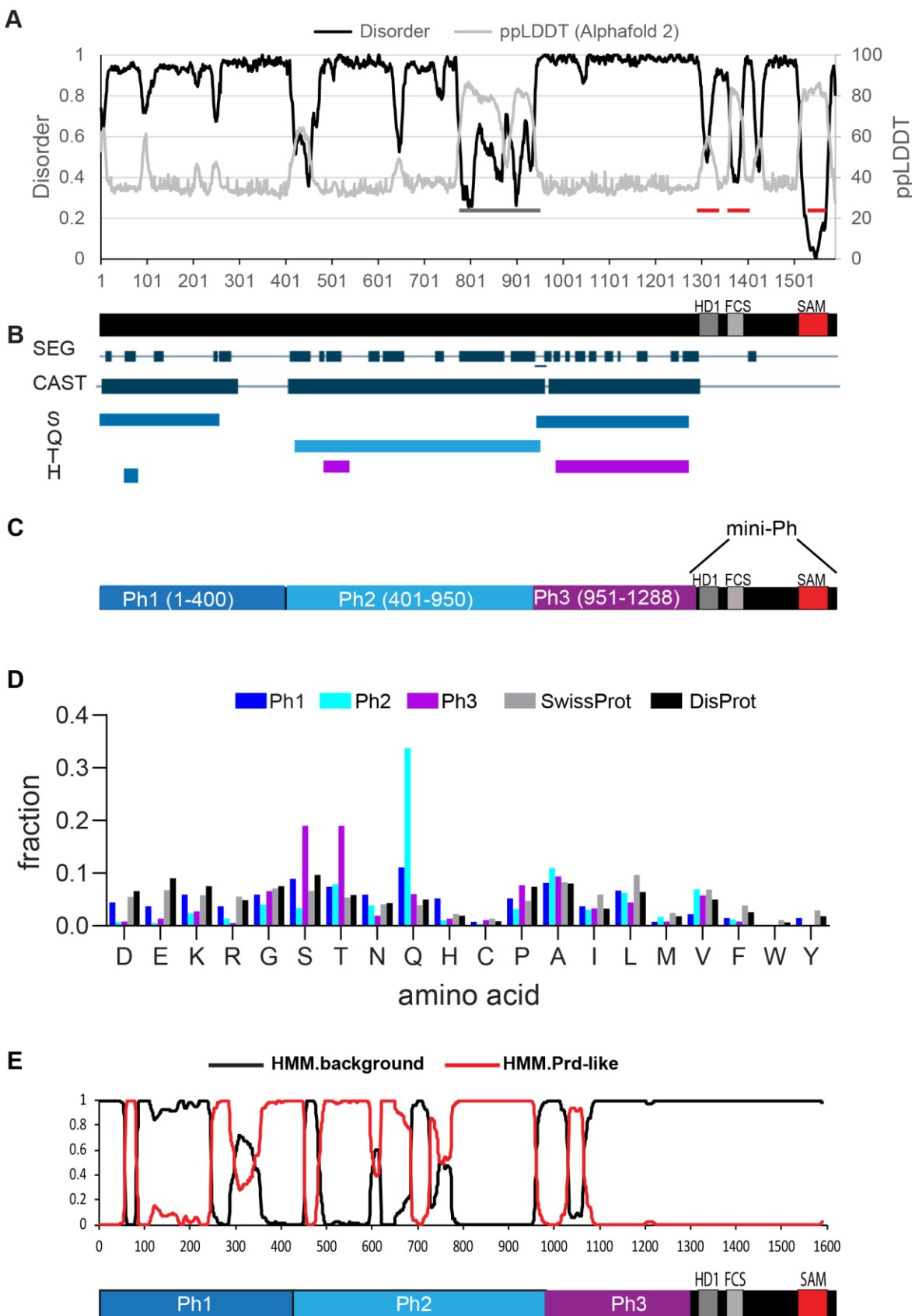

**Figure 1.** Identification of three IDRs in Ph. (**A**) Prediction of intrinsically disordered sequence using MetapredictV2, which integrates AlphaFold 2 predictions [32,33]. *x*-axis is amino acid position. Most

of Ph was predicted to be disordered (>0.5 on left *y*-axis), with the exception of Ph SAM (indicated with a red line). The two other red lines correspond to the HD1 and FCS motifs, which were scored as disordered, although the AlphaFold2 signal predicted some order. The grey bar indicates a region predicted to have some order; this is a densely Q-rich region that was predicted to form helices/coiled coils. (**B**) SEG/CAST analysis of Ph using PlaToLoCo [39] suggested that the disordered region could be parsed into subregions based on which amino acids were masked (repetitive). CAST track indicated all masked regions, and the masked amino acids are indicated below. (**C**) Schematic of different regions in the disordered Ph region defined as Ph1(IDR1), Ph2(IDR2) and Ph3(IDR3). (**D**) Frequency of each amino acid in the three IDRs relative to the SwissProt and DisProt databases. (**E**) Prion-like domain prediction using PLAAC [40]. IDR 2 was dominated by high scoring prion-like regions, but all three IDRs had predicted PrDs. See Figure S2 for the full sequence of each IDR.

## 2.2. Expression of Ph Proteins Lacking Each IDR, or Combinations of Them, in Cells

To understand the contribution of each IDR to Ph condensate formation, we designed a series of constructs consisting of N-terminal fusions of the Venus GFP variant to truncated versions of Ph. Constructs were designed to test: (1) the effect of removing each IDR from Ph, (2) the activity of each IDR when fused to mini-Ph (i.e., deletion of 2 IDRs), (3) the role of the SAM in IDR effects, and (4) the condensate-forming activity of each IDR alone or in combination. We used the heat shock promoter for inducible expression. Constructs were transiently transfected into *Drosophila* S2 cells, along with a plasmid encoding H2Av-RFP (under control of the Actin5C promoter) as a nuclear marker. Cells were subjected to mild heat shock (8 min at 37 °C) to induce protein expression, and were allowed to recover overnight before live imaging.

We first used Western blots to confirm that a full-length protein was expressed for each construct and that the expression levels were similar. There was a ~3× variability in the expression levels for different proteins, although transfection conditions were identical (Figure S3A–D). We also compared expression with endogenous Ph for proteins that maintained the antigen recognized by our antiserum (Figure S3E,F). Average total Ph levels did not exceed 2-fold that of untransfected cells. This may be due to the downregulation of endogenous *ph* by the transfected proteins. The *ph* locus is known to contain Polycomb response elements [41], which may mediate auto-regulation [42,43]. We cannot rule out other explanations for the reduced Ph levels, such as a response to transfection. Transfected cells also have a wide range of expression levels. Thus, although the global Ph levels were modestly increased at the population level, the analysis of individual cells by imaging indicated a wide range of expression levels (see below sections).

## 2.3. Analysis of the Function of Ph IDRs Using Live Imaging

As previously shown [26], transfected wild-type (WT) Ph forms several round, bright condensates in cells, while mini-Ph forms no condensates or a single condensate (Figure 2A–E). We carried out two analyses of the imaging data (see Methods for details). To analyze large numbers of cells in an unbiased manner, we developed Cell Profiler [44,45] analysis pipelines to identify condensates, count them, and measure their size. We also carried out a smaller manual analysis using ImageJ to count condensates and measure the total nuclear intensity of Venus. Because there are a range of expression levels in transfected cells and because we aimed to detect both small and large condensates, our images of very bright condensates sometimes contained saturated pixels. This resulted in the underestimation of intensities but should have effected measurements only from the highest expressing cells.

In vitro, mini-Ph forms condensates through phase separation, a process with a sharp concentration dependence. To assess the relationship between condensate formation and protein concentration, we plotted the mean intensities of at least 48 nuclei and coded them based on whether they had at least one condensate (Figure 2F). For each construct, we displayed 8 cells from the same image with increasing intensities (Figure 2B,D, cells 1–8). This analysis showed that Ph formed condensates even at the lowest concentrations we measured. In contrast, mini-Ph only formed condensates in cells with the highest intensities

(Figure 2D,F). This is consistent with the observation that removing the IDRs increased the threshold for condensate formation, although we cannot assess the threshold for Ph from our current data because of the low number of cells with detectable Venus-Ph expression that lacked condensates.

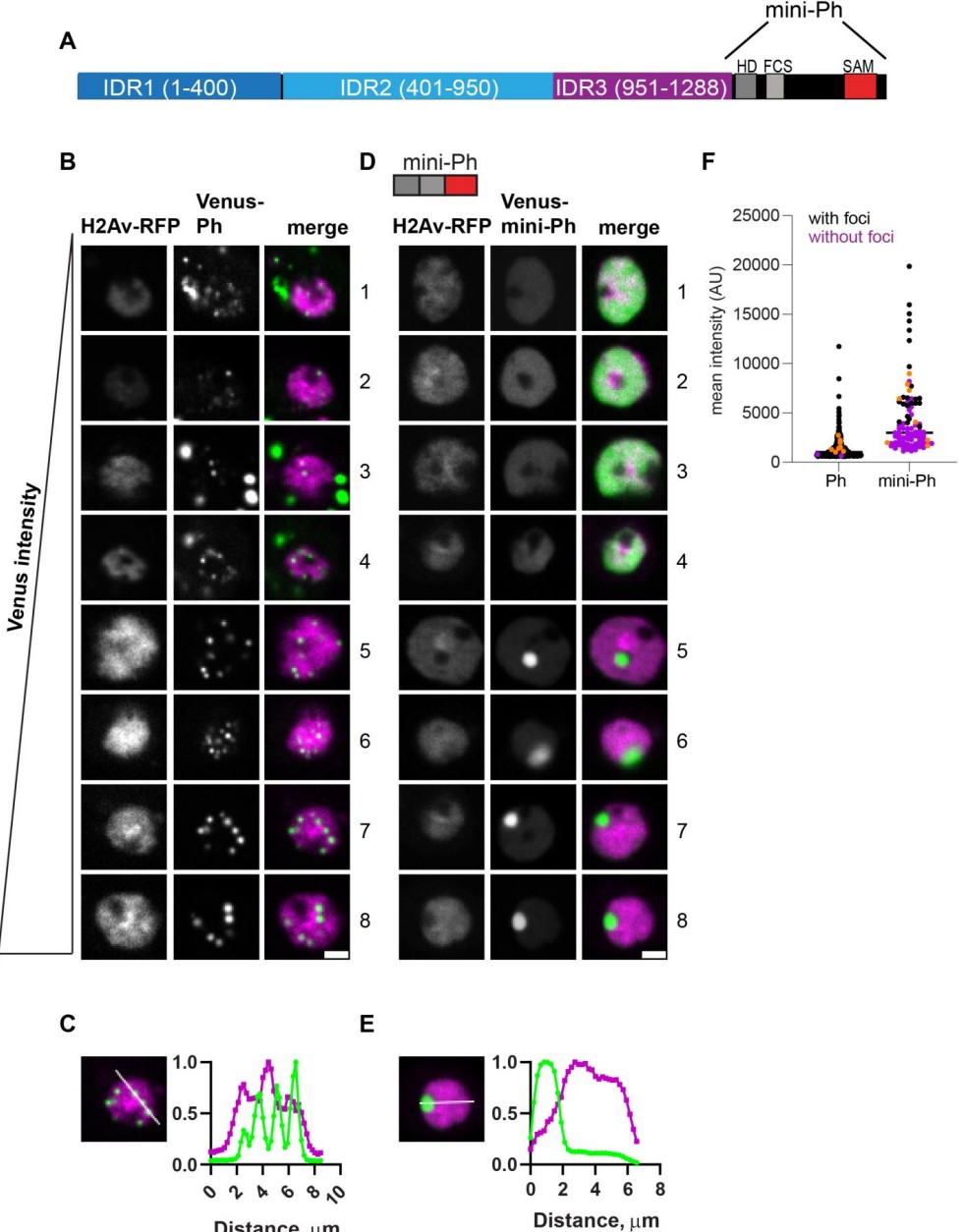

**Figure 2.** Characterization of Ph and mini-Ph condensates and their relationship to protein expression level. (**A**) Schematic of Polyhomeotic. (**B**–**E**) Representative images of live S2 cells that were co-transfected with Venus-Ph (**B**,**C**) or Venus-mini-Ph (**D**,**E**). H2Av-RFP was co-transfected as a nuclear marker. Images are from a single slice from a confocal stack and are arranged (1–8) based on the mean nuclear Venus intensity. Note that images were adjusted to make the signals visible for presentation, so the intensities cannot be compared across images. Scale bar is 3 microns. (**C**,**E**) Line scans through condensates to assess co-localization with chromatin. Both signals are scaled to their maximum intensity on the *y*-axis. (**F**) Relationship between mean nuclear Venus intensity and condensate formation. Cells without condensates are indicated in magenta, and those with condensates are indicated in black. Intensity values for the cells shown in (**B,D**) are indicated by orange symbols. Bar shows median. WT-Ph, *n* = 391; mini-Ph, *n* = 93.

Ph is a chromatin-bound protein. To determine whether condensates are likely to be chromatin-associated, we plotted the profile of line scans through large condensates formed in cells with the highest expression levels for both Ph and mini-Ph (Figure 2C,E). We found that while the small condensates formed by Ph overlapped with chromatin, the large mini-Ph condensates excluded chromatin (also clearly visible in images 5–8 in Figure 2D). Finally, we observed cytoplasmic condensates in ~30% of Ph-expressing cells (e.g., cells 1, 3, and 4 in Figure 2B), though we did not observe cytoplasmic condensates in cells expressing mini-Ph. Thus, removing the Ph IDRs increased the protein concentration required for nuclear condensate formation and led to the formation of large condensates that excluded chromatin. Mini-Ph, like Ph, was excluded from the nucleolus.

### 2.4. Removal of IDRs Affects Ph Condensate Size, Number, and Morphology

For each IDR, we compared the condensates formed when the IDR was deleted versus when it was the only IDR present.

The condensates that formed in the absence of Ph1 (PhΔ1) were heterogeneous, frequently small, and tended to form interconnected networks (Figure 3A). This likely explains the significant increase in foci size but not the number observed for PhΔ1 (Figure 3E,F). As PhΔ1 concentrations increased, condensates became more apparent, but substantial fluorescence outside condensates remained in most cells. The pattern of PhΔ1 outside of the condensates appeared granular, raising the possibility that PhΔ1 formed tiny clusters that could not be resolved with our imaging method. PhΔ1 condensates overlapped with chromatin (Figure 3B). Most cells with detectable PhΔ1 expression formed at least one visible condensate in our manual analysis, even when the nuclear intensity was low (Figure 3G).

The condensates formed by PhΔ2Δ3 (only Ph1 present) were completely different. Round condensates were easily visible at even the lowest protein concentrations (Figure 3C,G). At high protein concentrations, cells with very large condensates, reminiscent of mini-Ph-expressing cells, were observed (Figure 3C, cells 7, 8). For PhΔ2Δ3, the large condensates formed in cells with high protein levels clearly excluded chromatin (Figure 3D), and many smaller condensates appeared to do so as well (e.g., cell 4 in Figure 3C). We also noticed apparent chromatin "fissures" in some nuclei expressing PhΔ2Δ3 (arrows in Figure 3C). Condensates seemed to form or collect in these fissures. PhΔ2Δ3 cells formed significantly larger foci than WT-Ph cells (Figure 3F).

The expression of Ph lacking Ph2 (PhΔ2), the glutamine-rich IDR, resulted in a small number of large condensates (Figure 4A). Consistent with this impression, the number of foci per nucleus counted for PhΔ2 was reduced compared with the wild type and there was a significant increase in foci size (Figure 4E,F). These condensates were similar to those formed by mini-Ph, including that they clearly excluded chromatin (Figure 4B). However, cells with more than one mini-Ph condensate were very rare, and cells with two condensates were common for PhΔ2. PhΔ2-expressing cells also clearly had chromatin fissures (e.g., cells 5 and 7 in Figure 4A). Like mini-Ph, PhΔ2 had a high threshold for condensate formation, so that many cells with clear Venus-PhΔ2 expression were observed without condensates (Figure 4A,G).

The condensates formed by Ph containing only the Ph2 IDR (PhΔ1Δ3, Figure 4C) were completely distinct from PhΔ2 condensates. They were significantly smaller and more numerous than WT-Ph condensates (Figure 4E,F) and appeared to be embedded in chromatin (Figure 4D). Condensates formed even at the lowest expression levels analyzed, similar to WT-Ph (Figure 4G).

The condensates formed by Ph lacking the Ph3 IDR (PhΔ3) were bright and numerous, and formed at the lowest analyzed expression levels (Figure 5). As observed for WT-Ph, cytoplasmic condensates were frequently present in cells expressing PhΔ3 (37% cells; e.g., Figure 5A, cells 2 and 8). PhΔ3 condensates appeared most similar to WT-Ph condensates, although they were more numerous and (as a population) slightly larger (Figure 5E,F).

Nuclear PhΔ3 condensates overlapped with chromatin (Figure 5B) and formed at the lowest measured concentrations (Figure 5G).

When Ph3 was the only IDR present (PhΔ1Δ2), the condensates formed are similar to those formed by mini-Ph in that cells with high protein concentrations tended to form a single large condensate that excluded chromatin (Figure 5C,D). Cells expressing PhΔ1Δ2, even when it did not form condensates, had a heterogeneous chromatin distribution that appeared non-overlapping with PhΔ1Δ2 (Figure 5C). This pattern was not observed for mini-Ph or any of the other analyzed proteins. PhΔ1Δ2 showed a higher expression threshold for condensate formation, similar to mini-Ph (Figure 5G).

Taken together, the data indicate that each IDR affects condensate formation. The most obvious pattern was that proteins lacking Ph2 (PhΔ2, PhΔ2Δ3, and PhΔ1Δ2) consistently formed large round condensates that excluded chromatin, while those containing Ph2 formed small chromatin-associated condensates. Thus, Ph2 has the most dramatic effect on condensate formation. A comparison of single versus double IDR deletions also indicated that the IDRs can influence each other. Specifically, Ph3 inhibited the effects of both Ph1 and Ph2. Ph1 promoted the formation of small round condensates, though only in the absence of Ph3 (compare PhΔ2Δ3, Figure 3C with PhΔ2, Figure 4A). Ph2 drove the formation of small condensates at low concentrations, but this was more evident in PhΔ1Δ3 than in PhΔ1 (compare Figure 3A with Figure 4C). Figure S4 compiles results from all proteins to facilitate comparisons among the effects of the IDRs.

*2.5. Ph IDRs Regulate Chromatin Association*

To understand how the Ph IDRs affect chromatin association, we used subcellular fractionation. Transfected cells expressing Venus-tagged Ph proteins cells were fractionated into cytoplasmic, soluble nuclear, and chromatin-associated fractions (Figure 6 and Figure S5). As expected, and consistent with previous (and current) analyses of endogenous Ph [46] (Figure 6B), Venus-Ph was mainly present in the chromatin and soluble nuclear fractions (Figure 6C,D). The removal of the SAM, which is required for condensate formation by full length Ph, increased the fraction of protein in the soluble nuclear fraction (Figure 6D). Analysis of Ph lacking IDRs indicated that proteins lacking Ph2 (mini-Ph, PhΔ2, PhΔ1Δ2, and PhΔ2Δ3) were mainly present in the cytoplasmic (mini-Ph and PhΔ2) and soluble nuclear (PhΔ1Δ1 and PhΔ2Δ3) fractions, with reduced levels on chromatin (Figure 6E). This is consistent with the observed chromatin exclusion in the microscopy experiments, although we did not clearly observe cytoplasmic protein via microscopy. It is possible that the protein detected in the cytoplasm was released during the fractionation. Proteins containing Ph2 but lacking one or two IDRs (PhΔ1, PhΔ3, and PhΔ1Δ3) were more strongly chromatin-associated than Ph, consistent with the apparent overlap of these condensates with chromatin (Figure 6D). The removal of Ph1, Ph3, or both increased the fraction of chromatin-bound Ph, which was consistent with the observation that Ph3 and Ph1 inhibited chromatin association mediated by Ph2. The fractionation results are summarized in Figure 6F, which clearly shows how Ph2 alters the distribution of Ph in the cell.

*2.6. Removal of the Ph3 IDR Allows Ph2-Dependent but SAM-Independent Condensates to Form*

We previously showed that Ph SAM is necessary for condensates to form in cells [26]. To determine if the Ph IDR deletion constructs formed condensates in the absence of the SAM, we tested all the constructs except PhΔ1Δ2 without the SAM (Figure 7A). In the absence of the SAM, most proteins did not form condensates. However, in two cases, the deletion of Ph3 and the deletions of both Ph1 and Ph3, condensates formed without the SAM, although the number of cells that formed condensates was low (Figure 7B,C). Although the ΔSAM versions were expressed at about 2-fold lower levels than the corresponding proteins with the SAM (Figure S3), given the wide range of expression levels over which condensates were observed, this is unlikely to explain why most ΔSAM proteins did not form condensates. The ΔSAM versions were also expressed at similar levels as transfected WT-Ph (Figure S3).

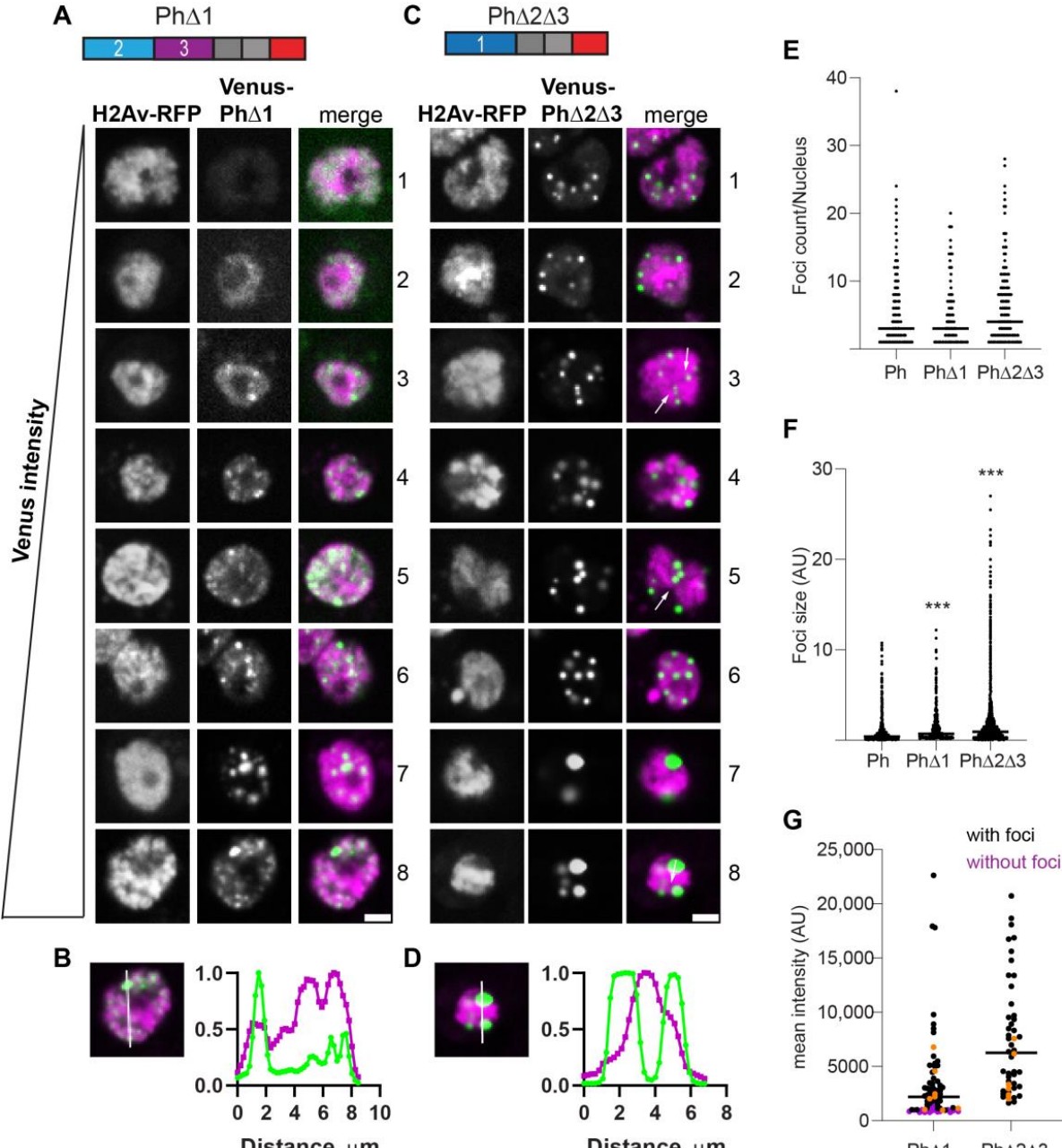

**Figure 3.** Effect of Ph1 IDR on condensates. (**A–D**) Representative images of live S2 cells that were co-transfected with Venus-PhΔ1 (**A,B**) or Venus-PhΔ2Δ3 (**C,D**). H2Av-RFP was co-transfected as a nuclear marker. Images are from a single slice from a confocal stack and are arranged based on the mean nuclear Venus intensity. Images were adjusted to make the signals visible for presentation, so the intensities cannot be compared across images. Arrows indicate chromatin "fissures". Scale bar is 3 microns. (**B,D**) Line scans through condensates to assess co-localization with chromatin. Both signals are scaled to their maximum intensity (*y*-axis). (**E,F**) Graph of the number of condensates (foci) per nucleus (**E**) and condensate size (**F**) from Cell Profiler analysis. The total number of transfected cells analyzed (with or without foci): WT-Ph, *n* = 3478; PhΔ1, *n* = 2426; PhΔ2Δ3, *n* = 2911. *p*-values are presented for comparison with WT using the Kruskal–Wallis test with Dunnett's correction for multiple comparisons. *** $p < 0.0001$. (**G**) Relationship between mean nuclear intensity and condensate formation. Cells without condensates are indicated in magenta, and those with condensates are indicated in black. Intensity values for the cells shown in (**A,C**) are indicated by orange symbols. Bar shows median. PhΔ1, *n* = 78; PhΔ2Δ3, *n* = 48.

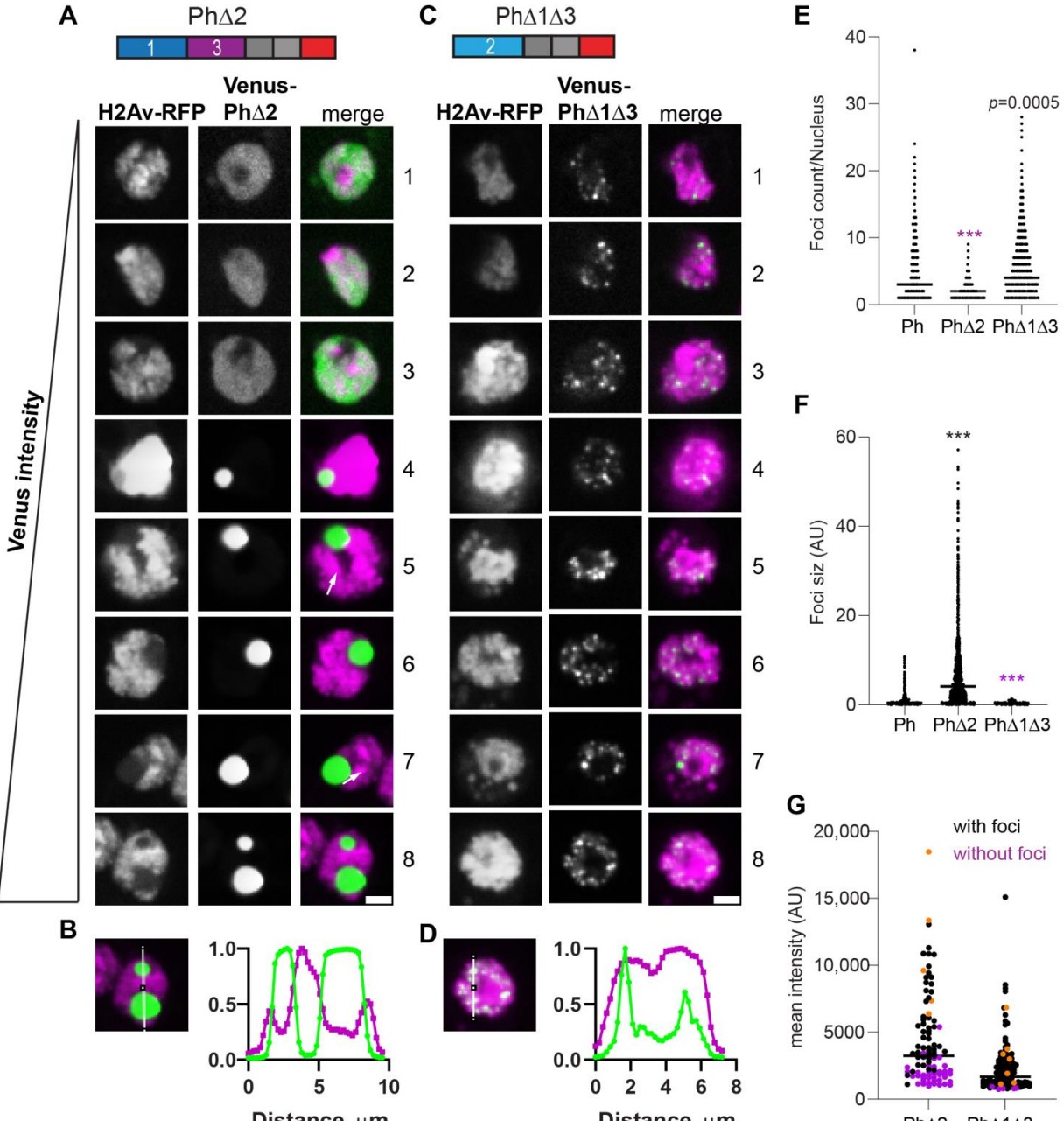

**Figure 4.** Effect of Ph2 IDR on condensates. (**A–D**) Representative images of live S2 cells that were co-transfected with Venus-PhΔ2 (**B**) or Venus-PhΔ1Δ3. H2Av-RFP was co-transfected as a nuclear marker. Images are from a single slice from a confocal stack and are arranged based on the mean nuclear Venus intensity. Images were adjusted to make the signals visible for presentation, so the intensities cannot be compared across images. White arrows indicate examples of chromatin "fissures". Scale bar is 3 microns. (**B,D**) Line scans through condensates to assess co-localization with chromatin. Both signals are scaled to their maximum intensity (*y*-axis). (**E,F**) Graph of the number of condensates (foci) per nucleus (**E**) and condensate size (**F**) from Cell Profiler analysis. The total number of transfected cells analyzed (with or without foci): WT-Ph, *n* = 3478; PhΔ2, *n* = 1539; PhΔ1Δ3, *n* = 1568. *p*-values are presented for comparison with WT using the Kruskal–Wallis test with Dunnett's correction for multiple comparisons. *** *p* < 0.0001. (**G**) Relationship between mean intensity and condensate formation. Cells without condensates are indicated in magenta, and those with condensates are indicated in black. Intensity values for the cells shown in (**A,C**) are indicated by orange symbols. Bar shows median. PhΔ2, *n* = 95; PhΔ1Δ3, *n* = 124.

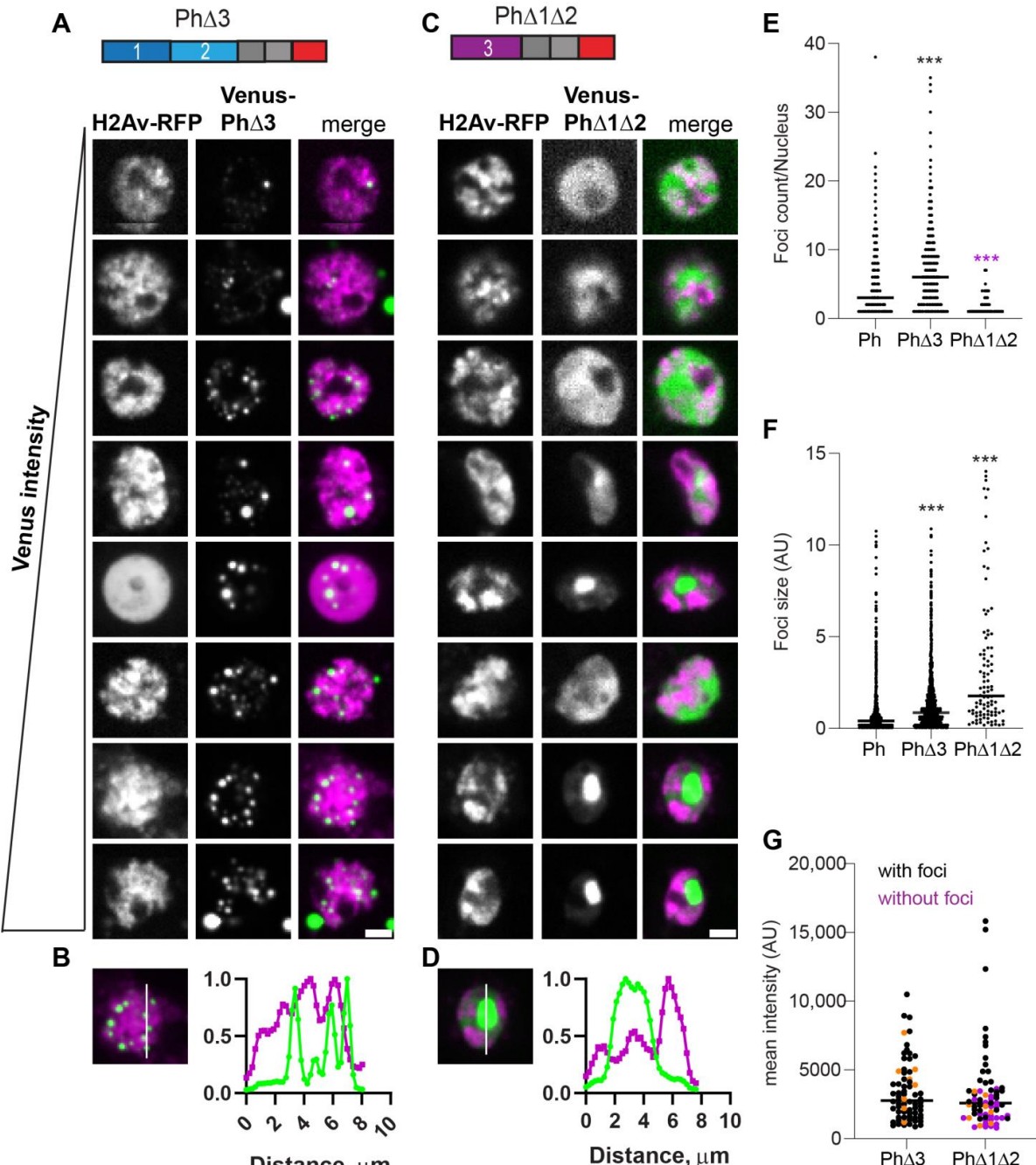

**Figure 5.** Effect of Ph3 IDR on condensates. (**A–D**) Representative images of live S2 cells that were co-transfected with Venus-PhΔ3 (**B**) or Venus-PhΔ1Δ2. H2Av-RFP was co-transfected as a nuclear marker. Images are from a single slice from a confocal stack and are arranged based on the mean nuclear Venus intensity. Images were adjusted to make the signals visible for presentation, so the intensities cannot be compared across images. Scale bar is 3 microns. (**B,D**) Line scans through condensates to assess co-localization with chromatin. Both signals are scaled to their maximum intensity (*y*-axis). (**E,F**) Graph of the number of condensates (foci) per nucleus (**E**) and condensate size (**F**) from Cell Profiler analysis. The total number of transfected cells analyzed (with or without foci): WT-Ph, *n* = 3478; PhΔ3, *n* = 1557; PhΔ1Δ2, *n* = 1038. *p*-values are presented for comparison with WT using the Kruskal–Wallis test with Dunnett's correction for multiple comparisons. *** *p* < 0.0001 (**G**) Relationship between mean intensity and condensate formation. Cells without condensates are indicated in magenta, and those with condensates are indicated in black. Intensity values for the cells shown in (**A,C**) are indicated by orange symbols. Bar shows median. PhΔ3, *n* = 71; PhΔ1Δ2, *n* = 67.

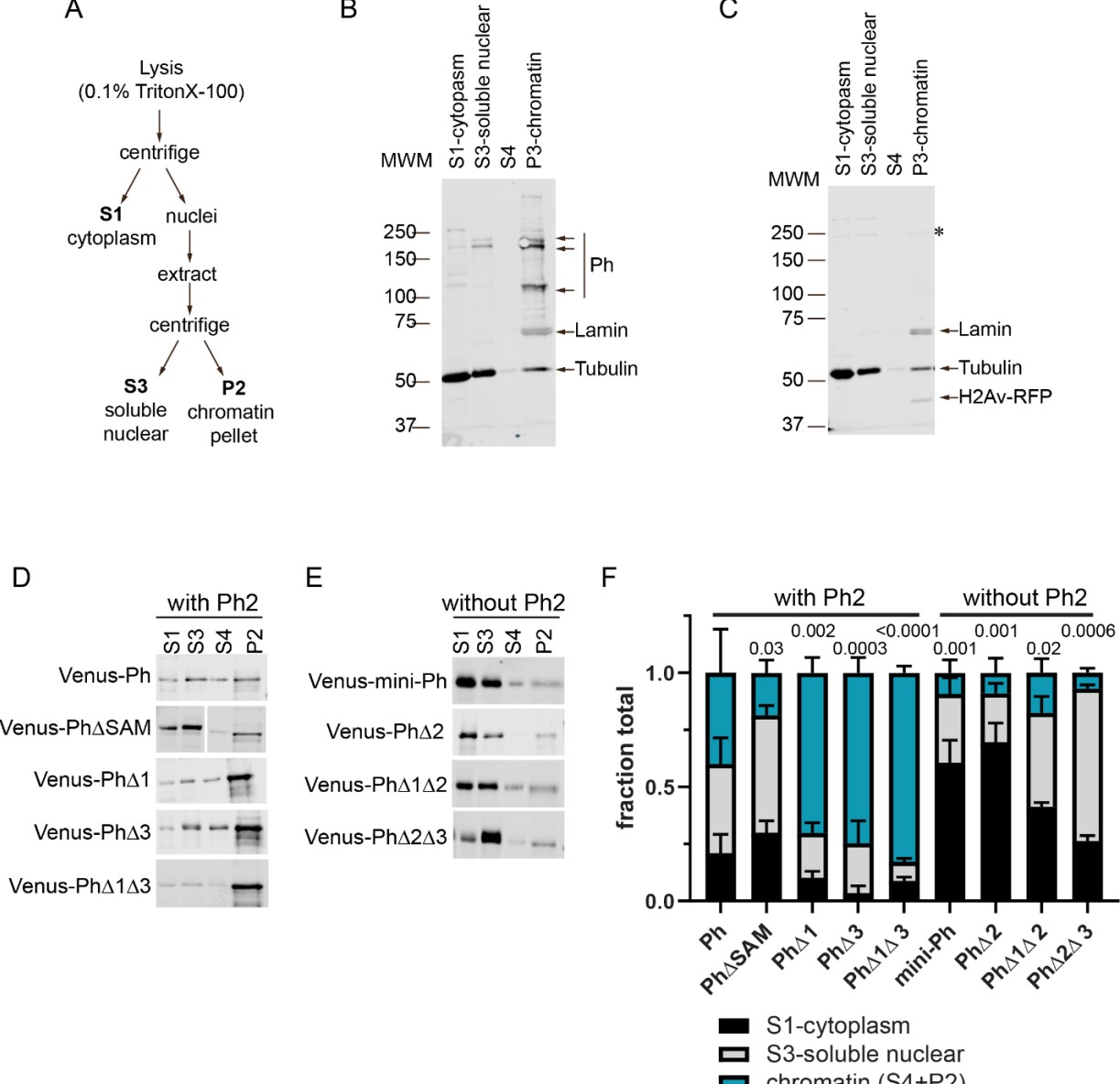

**Figure 6.** Effect of Ph IDRs on chromatin association. (**A**) Schematic of cell fractionation protocol (based on [47]). Note that P2 was digested with DNaseI and RNaseA, and the supernatant from this was analyzed as S4 on blots. However, this digest was not successful because histones were not released into the supernatant, so we pooled the signals from S4 and P2 for quantification. (**B**) Representative Western blot of the subcellular fractionation of untransfected S2 cells. Lamin was used as a nuclear marker, and tubulin was used as a cytoplasmic one. Endogenous Ph (Ph-p and Ph-d, with major isoforms at ~170 kDa and ~150 kDa) was mainly found in the chromatin fraction. (**C**) Representative Western blot of the subcellular fractionation of S2 cells transfected with Venus-Ph. Blots were probed with anti-GFP to recognize Venus-Ph (faintly visible, asterisk) and reprobed with anti-tubulin, anti-lamin, and anti-RFP to detect co-transfected H2Av-RFP. (**D,E**) Representative Western blots of the fractions of cells transfected with the indicated constructs, separated by whether they included (**D**) or did not include (**E**) the Ph2 IDR. Venus-Ph proteins were detected with anti-GFP. Note that in all Western blots, 2× more P2 was loaded than other fractions. Fractions for cells transfected with constructs lacking Ph2 were loaded at 0.67× the amount of those expressing constructs with Ph2. (**F**) Summary of quantification of three independent fractionation experiments. Error bars show mean and standard deviation. Numbers are *p*-values for one-way ANOVA comparing the fraction in chromatin relative to WT-Ph.

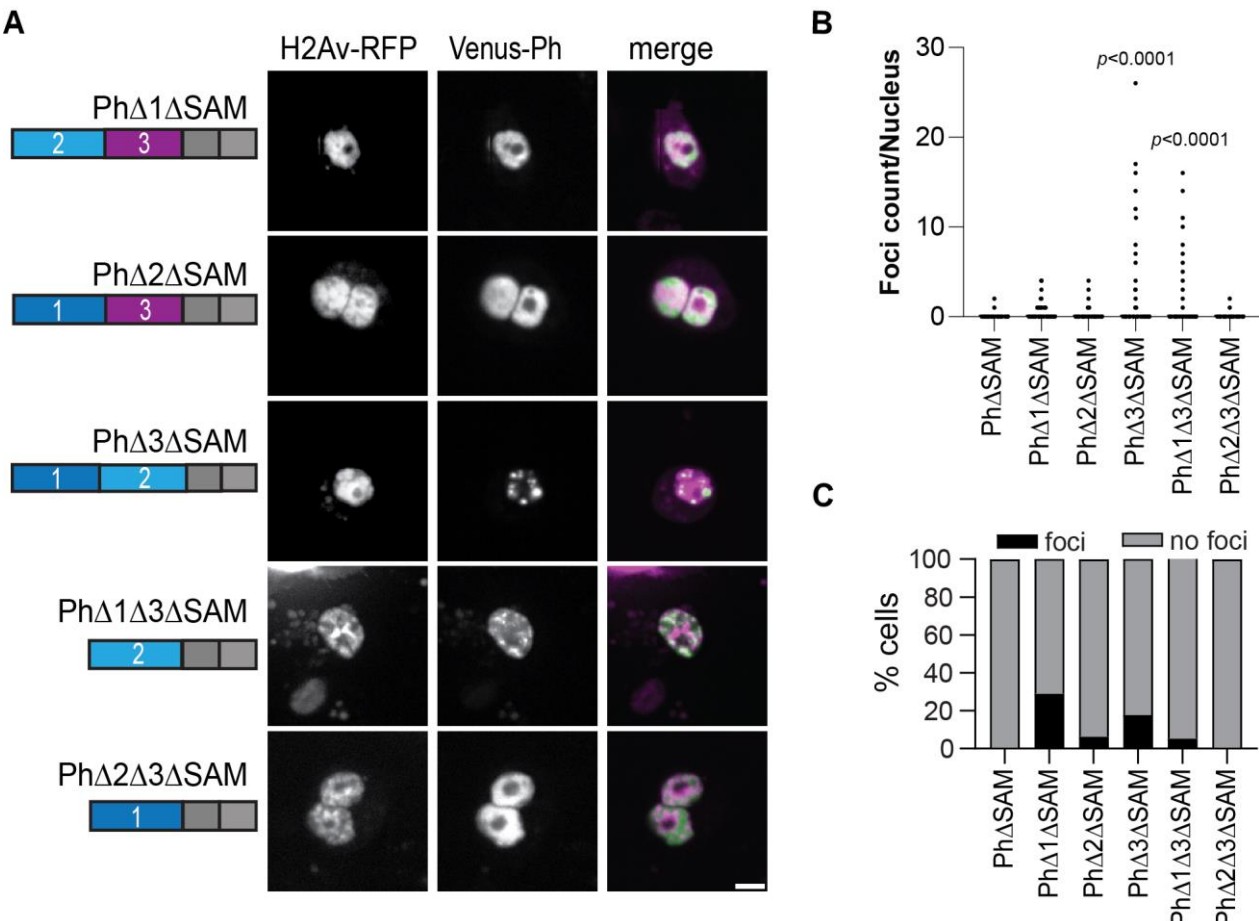

**Figure 7.** Effect of Ph IDRs on condensate formation in the absence of the SAM. (**A**) Representative live images of S2 cells transfected with Venus-Ph constructs lacking one or two IDRs and the SAM with H2Av-RFP as a nuclear marker. Images show maximum intensity projections of confocal stacks. Scale bar is 5 microns. (**B**) Quantification of foci count per nucleus for H2Av-RFP-positive cells. (**C**) Percent of H2Av-RFP-positive cells that formed foci. Note that for PhΔ1ΔSAM, most of the foci-forming cells were scored based on a single condensate. Manual inspection suggested that this likely reflected uneven protein distribution rather than true condensates. At least 100 cells from each of the three independent experiments were analyzed. The total number of cells analyzed was: PhΔSAM, $n = 1348$; PhΔ1ΔSAM, $n = 411$; PhΔ2ΔSAM, $n = 359$; PhΔ3ΔSAM, $n = 214$; PhΔ1Δ3ΔSAM, $n = 922$; PhΔ2Δ3ΔSAM, $n = 1413$. *p*-values are presented for comparison with PhΔSAM using the Kruskal–Wallis test with Dunnett's correction for multiple comparisons.

### 2.7. Ph IDRs Alone and in Combination Can Form Condensates

To test if the IDRs could form condensates in the absence of mini-Ph, we tested each IDR alone or in combination (Figure 8). An SV40 NLS was added to ensure that the IDRs localized to the nucleus, since the Ph NLS is in the FCS domain. The proteins were expressed within 2× the level of transfected WT-Ph, and total Ph levels were not increased more than 2× (Figure S6). Ph3 alone did not form condensates. However, both Ph 1 and Ph2 formed condensates in a very small number of cells (Figure 8B). When Ph1 and Ph2 were combined (Ph5), many small condensates were formed in many cells (Figure 8B,C). In contrast, when Ph2 and Ph3 were combined (Ph6), condensates did not form. Finally, when Ph1, Ph2, and Ph3 were combined (Ph7), condensates were formed, although fewer than with Ph1 and Ph2 (Ph5) (Figure 8B,C). These results indicate that the intrinsic activity of the IDRs may contribute to Ph condensates but may also be constrained by the mini-Ph region. The finding that Ph5 frequently formed condensates, Ph6 did not, and Ph7 formed

fewer confirms the inhibitory effect of Ph3 on Ph1 and Ph2. Table 1 shows a comprehensive summary of condensate formation for all of the tested constructs.

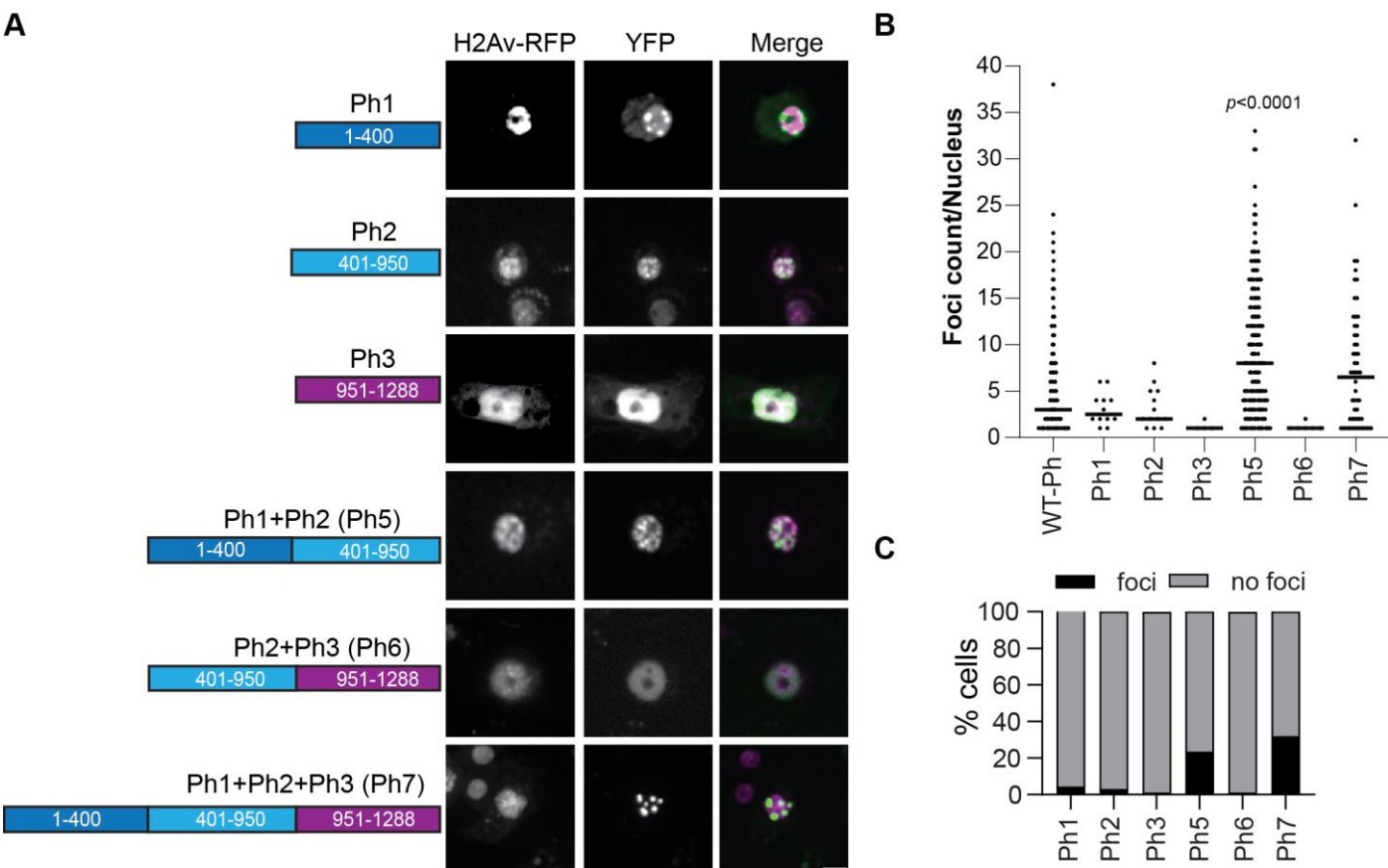

**Figure 8.** Condensate formation by Ph IDRs. (**A**) Representative live images of S2 cells transfected with the Venus-Ph truncation constructs (IDRs only) with H2Av-RFP as a nuclear marker. Images show maximum intensity projections of confocal stacks, and the scale bar is 5 microns. (**B**) Quantification of foci per nucleus. Bars show the median value for cells that formed foci, pooled from three independent experiments. (**C**) Percent of H2Av-RFP-positive cells that formed foci. The total number of transfected cells analyzed (with and without foci) were as follows: Ph1, *n* = 268; Ph2, *n* = 412; Ph3, *n* = 309; Ph5, *n* = 857; Ph6, *n* = 460; Ph7, *n* = 175. *p*-values are presented for comparison with WT-Ph using the Kruskal–Wallis test with Dunnett's correction for multiple comparisons.

**Table 1.** Summary of condensate formation by Ph proteins.

| Construct | SAM | Condensates | Condensate Size |
|---|---|---|---|
| PhΔ1 | + | ++ | small |
| PhΔ2 | + | + | large |
| PhΔ3 | + | +++ | small |
| PhΔ1Δ3 | + | +++ | small |
| PhΔ2Δ3 | + | ++ | small and large |
| PhΔ1Δ2 | + | + | large |
| PhΔ1ΔSAM | - | - | - |
| PhΔ2ΔSAM | - | - | - |
| PhΔ3ΔSAM | - | ++ | small |
| PhΔ1Δ3ΔSAM | - | ++ | small |

In summary, we identified three IDRs in the N-terminal region of Ph, each of which influences Ph SAM-dependent condensates in cells. The most dramatic effects occurred

when the glutamine-rich Ph2 IDR was removed, which reduced chromatin binding and led to the formation of large, round condensates. The Ph3 IDR, which was previously shown to inhibit the aggregation of the SAM through its O-linked glycosylation, also inhibited Ph1 and Ph2 activities. The effects of the IDRs correlated with their effects on chromatin association—weak chromatin association corresponded to large, round condensates, while strong chromatin association correlated with small, chromatin-associated condensates. Figure 9 summarizes these results.

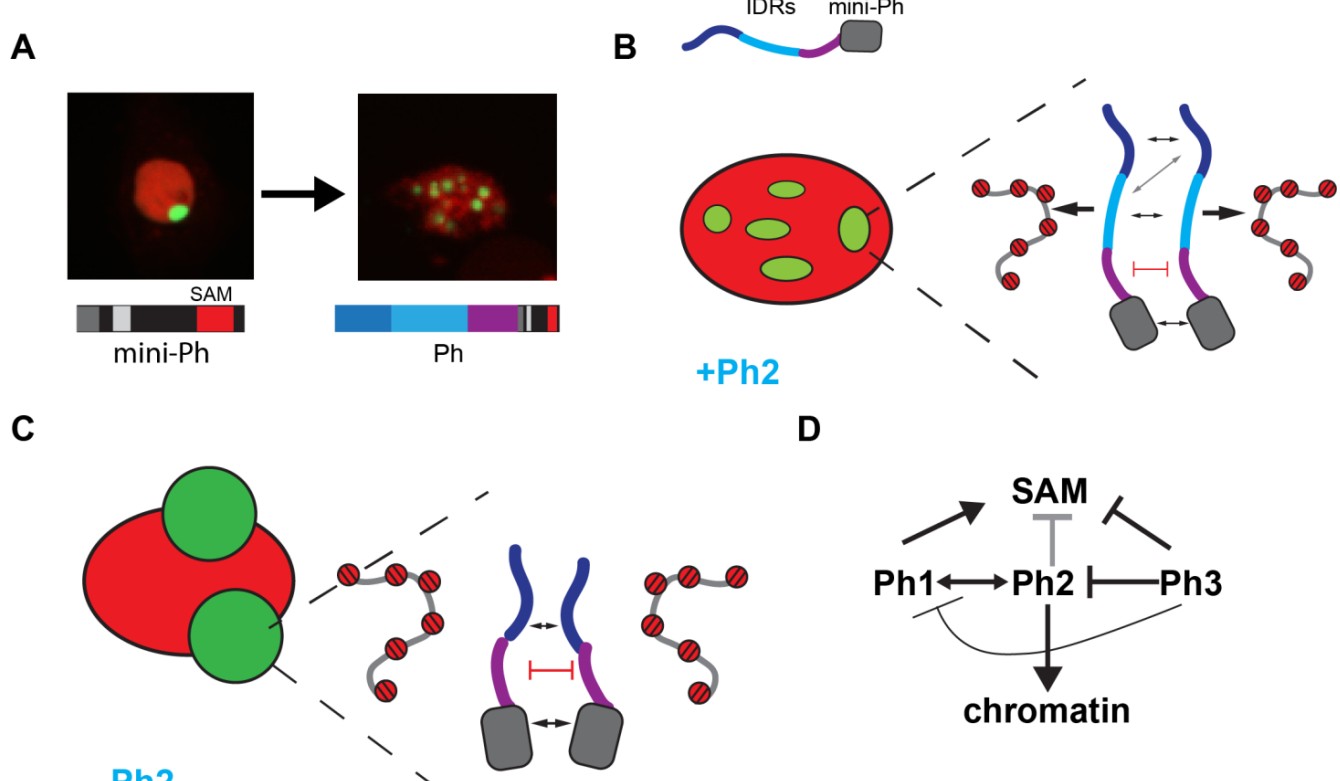

**Figure 9.** Model for role of Ph IDRs in SAM-dependent condensate formation. (**A**) Comparison of condensates formed by mini-Ph and Ph demonstrates the effect of the IDRs. (**B**) Schematic of interactions that may be balanced to form multiple condensates in constructs containing the Ph2 IDR. Protein–protein interactions between the mini-Ph region and among the IDRs are predicted to drive phase separation, while tight binding to chromatin mediated by Ph2 may restrict condensate formation. (**C**) Schematic of how removing Ph2 may lead to the formation of large condensates by removing the constraint imposed by tight chromatin binding. (**D**) Working model for regulatory network among IDRs. Ph2 (gray arrow) restricts both the size of condensates and propensity for formation when Ph SAM is present, and it imparts tight chromatin association. Ph3 inhibits the ability of both Ph1 and Ph2 to promote condensate formation; it may also inhibit the SAM (as previously proposed).

## 3. Discussion

We sought to determine how SAM-dependent condensates that may form by phase separation are controlled by the disordered N-terminal region of Ph. We identified three distinct IDRs in this region by sequence analysis and tested the function of each, in the context of full length Ph or Ph without the SAM, and alone. We found that each IDR affected condensates and binding to chromatin and that the IDRs functionally interacted with each other. The potential mechanisms underlying these effects are considered below.

Ph1 was found to promote condensate formation. When Ph1 was fused to mini-Ph (PhΔ2Δ3), small round condensates were formed that were wild-type-like (although more numerous than those formed by WT-Ph) (Figure 3). These condensates were dependent

on Ph SAM (Figure 7). When Ph1 was deleted, condensates were less round, and diffuse granular staining was often observed. Ph1 alone could form condensates in cells, although it did so rarely (Figure 8). Taken together, these data suggest that Ph1 promotes the condensate-forming ability of Ph SAM, leading to the formation of many condensates in a high fraction of cells.

Ph2 had the most profound effects on condensates. Removal of Ph2 led to the formation of very large condensates, similar to those formed by mini-Ph (Figure 4). When Ph2 was fused to mini-Ph, many small condensates were formed (although in a small number of cells). Ph2 promoted condensate formation in the absence of the SAM, but only if Ph3 was absent (Figure 7). Ph2 alone had a weak condensate-forming ability, but when fused to Ph1, formed a large number of small condensates (Figure 8). Thus, Ph2 negatively regulates Ph SAM-dependent condensates but also functionally interacts with the other IDRs in a SAM-independent manner—it synergizes with Ph1 to promote condensates and is inhibited by Ph3. The most striking sequence features of Ph2 are its high glutamine (36%) content and presence of prion-domains (Figure 1). It is also intriguing to note that a frequently used *ph* allele, *ph*^{505}, which was thought to be a null, actually encodes truncated versions of both *ph-p* and *ph-d*. The predicted proteins are truncated midway through the Ph2 region, so they could potentially produce proteins similar to Ph5 (i.e., Ph 1+2). The effects of *ph*^{505} on gene regulation slightly differ from those of a true *ph* null [48], raising the possibility that the effects of *ph*^{505} are due to expression of a Ph5-like protein. Although we refer to Ph2 as an IDR, its glutamine-rich regions are predicted to have helix-forming potential. Thus, future studies should assess a possible contribution of helix-formation in the activity of Ph2, particularly in light of recent work implicating the helix-forming potential of glutamine-rich sequences in regulating phase separation [37].

Ph3 has inhibitory effect on condensates, so that more condensates formed when it was removed. Careful consideration of the effects of removing and having Ph3 in different constructs suggests that Ph3 may function through the other IDRs, by restricting the activity of Ph2. First, when Ph3 was fused to mini-Ph (PhΔ1Δ2), few condensates were formed, and the ones that formed were similar to those formed by mini-Ph alone, suggesting that Ph3 had little effect in this context (although it appeared to globally affect chromatin organization) (Figure 5). Second, Ph1 fused to mini-Ph (PhΔ2Δ3) formed several condensates (Figure 3), but the number was significantly reduced (and size increased) when Ph3 was present with Ph1 (PhΔ2) (Figure 4). Third, Ph2 formed condensates in the absence of the SAM (PhΔ1Δ3ΔSAM) but not in the presence of Ph3 (PhΔ1ΔSAM) (Figure 7). Fourth, Ph2 alone formed condensates in some cells, but condensate formation was again inhibited when Ph2 was fused to Ph3 (Ph6) (Figure 8). Thus, Ph3 may function through the other IDRs rather than directly on Ph SAM; alternatively, it could inhibit both Ph SAM and Ph2. Ph3 is rich in serine and threonine, and is extensively modified by O-linked glycosylation of these residues [31], as well as by phosphorylation [49]. Removing glycosylation drives the formation of Ph aggregates in vitro and in vivo, and these aggregates depend on the SAM [25]. Whether these aggregates also depend on Ph2 is not known. However, one hypothesis would be that glycosylation restrains the activity of Ph2, which promotes condensate formation through SAM-dependent and -independent mechanisms. Recently, O-GlcNacylation was also shown to reduce both the aggregation and phase separation of the N-terminal LCR of the EWS protein [50]; future experiments will test if this is also true for Ph. It should be noted that a protein very similar to PhΔ3 was expressed from a transgene in *Drosophila* imaginal discs and formed condensates similar to wild-type Ph, consistent with our observations in S2 cells [25].

The SAM domain is not unique to Ph, but it is found in proteins of diverse function, from transcription factors to membrane receptors [51]. Many, but not all, SAMs are able to form polymers [52]. Two other PcG proteins, Sfmbt and SCM, have SAM domains. Sfmbt is part of a PcG complex called Pho-RC, which is important for targeting PcG proteins to specific DNA sequences recognized by the Pho subunit [53,54]. SCM strongly interacts with PRC1 through SAM–SAM interactions between Ph and SCM [55,56], and it can also

interact with PRC2 [57]. The SAM of SCM can co-polymerize with Ph [58], and can also interact with the SAM of Sfmbt, allowing a "tri-SAM" network to form (Ph–SCM–Sfmbt). SCM and Sfmbt also synergize to repress transcription [59–61]. Thus, heterotypic SAM interactions may integrate PcG complexes. How heterotypic SAM interactions contribute to condensate formation, including in our experiments where Ph was overexpressed, is not known. However, it is possible that SCM and Sfmbt can contribute to Ph behavior, even when Ph is present at much higher levels, for example, if SCM interacts with a single Sfmbt and a Ph oligomer.

We observed cytoplasmic condensates of WT-Ph and PhΔ3 in a high fraction of cells. Cytoplasmic condensates have not been reported in studies of GFP-Ph expressed in embryos [62], and we did not observe them in untransfected formaldehyde-fixed cells immunostained for anti-Ph. It is thus likely that cytoplasmic condensates form when Ph is present at high levels. This demonstrates that Ph condensates can form without binding to chromatin, although we do not know if the cytoplasmic and nuclear condensates have similar properties. It is possible that Ph and PhΔ3, both of which have strong condensate-forming activity, rapidly form condensates in the cytoplasm and that condensate formation interferes with nuclear transport. Cytoplasmic condensates could also reflect the nuclear shuttling of Ph, which has not been described, or a cytoplasmic function for Ph. Each of these possibilities could be tested in future experiments.

Many mechanisms have been described and hypothesized to explain the control of condensate size and number in cells [63–65]. One model that may be relevant for Ph1 and Ph2 explains how chromatin binding affects condensate number and size. Qi and Zhang used simulations to show that protein–chromatin interactions can promote a multi-droplet state [65]. In this model, protein–chromatin contacts promote the nucleation of phase separation but inhibit their coalescence. A detailed investigation of the thermodynamics underlying these effects indicated that this is due to a kinetic barrier between single- and multi-droplet states. This kinetic barrier arises from the chromatin network—when droplets that interact with chromatin coalesce, the chromatin network is constrained, which becomes progressively more energetically costly. Theoretical work from Wingreen and colleagues also showed that chromatin interferes with the coarsening of condensates, both by acting as a crowder and through its crosslinked network [64]. Thus, changing protein–chromatin interactions could change droplet number. In the framework of these models, Ph1 may decrease chromatin-binding affinity, promoting the formation of many round condensates. The loss of Ph1 may lead to tight chromatin binding, which would explain the distorted shape of the condensates formed by PhΔ1 and be consistent with our chromatin fractionation data. Ph2, on the other hand, promotes chromatin binding so that its loss leads to large condensates because fusion is not impaired.

Shin et al. [66] analyzed condensate formation by chromatin proteins using optogenetic manipulation to induce clustering/phase separation. They found that induced condensates of different chromatin proteins, as well as endogenous condensate-forming nuclear proteins, occupy regions of low chromatin density. This result is consistent with what we observed with large Ph condensates. Shin et al. [66] also showed that although bulk chromatin is excluded from condensates, specific chromatin regions that interact with the phase-separating protein associate with condensates. These observations lead to a model where condensates can act as a filter to bring specific chromatin regions together while excluding bulk chromatin [66]. Thus, an important next step will be to determine how the Ph IDRs affect Ph binding to specific PcG targets in chromatin (by chromatin immunoprecipitation). Although several TFs are implicated in PcG targeting, in most cases, the precise involved sequences have not been determined. Furthermore, the IDRs in TFs have been implicated in selection of which cognate DNA motifs are actually bound [67]; it is thus possible that the Ph IDRs participate in both the targeting and function of Ph. The experiments here motivate the further study of the Ph IDRs using physiological expression levels, as well as the in vitro analysis of the molecular basis of their effects on condensates.

Recently, Seydoux and colleagues demonstrated that clusters of MEG-3 function as Pickering particles, which adsorb to the surface of P-granules and reduce their coarsening by reducing surface tension [68]. As speculated, RNA may also have this function for some condensates [68]. In this regard, it is interesting that the Ph2 region was predicted (using DisoRDPbind [69]) to have a strong RNA-binding activity, although this has not been experimentally validated.

A caveat to this work is that we studied the behavior of Ph proteins under conditions of overexpression and with endogenous Ph present in the background. While this allowed us to evaluate how the different IDRs affected Ph condensate formation in a cellular context, we do not know which effects are dependent on endogenous Ph. The total Ph levels of transfected cells were not more than 2.5 fold higher than of untransfected cells due to the downregulation of endogenous Ph (Figure S3E,F). All of the constructs except the Ph IDRs alone (Figure 8) should assemble into PRC1, since this is mediated by the HD1 domain. However, our experiments do not determine which IDR effects are intrinsic to the IDR sequences versus due to interactions with cellular proteins. These interactions may be saturated in cells with high expression of the ectopic Ph proteins, but overexpression can also lead to spurious interactions. Our experiments revealed the complex effects of the IDRs; determining how these properties affect Ph and its function in gene regulation under normal physiological conditions requires additional experiments. This cell-based characterization of the Ph IDRs also motivates future in vitro and in silico analysis to understand which sequence properties drive the effects of the IDRs and how activities such as DNA/chromatin (or RNA) binding and phase separation contribute. We previously showed that mini-Ph undergoes phase separation in vitro; some of the condensates observed here (particularly large ones) were round and could be observed to fuse, properties consistent with (but not diagnostic of) phase separation. However, whether the small condensates driven by the presence of Ph2 form through a different mechanism awaits further study.

In conclusion, we showed that the disordered N-terminal region of Ph, which was largely uncharacterized, affects Ph SAM-dependent condensate formation in cells, likely by affecting Ph–chromatin interactions. The complex effects of the IDRs on condensates, and the functional interactions among them, indicate that condensate formation by the PcG is a tightly regulated process that integrates IDRs and structured regions such as the SAM. Most isoforms of all three mammalian Ph homologues have disordered N-terminal IDRs; we hypothesize that the IDR regulation of condensate formation by these IDRs may be functionally conserved.

## 4. Materials and Methods

Cloning: Ph and all of its truncations and deletions were cloned into a gateway donor vector using restriction digest and ligation, PCR, or gene blocks spanning the deletion junctions. No extra sequence was added—for example, PhD2 consists of aa1-400-951-1589. All synthetic and PCR-generated sequences were confirmed by Sanger sequencing. The plasmid for Ph truncations (pCR8-ATG-NLS) includes an SV40 nuclear localization sequence (NLS). To transfer Ph truncations into plasmid pHVW (*Drosophila* Genomics Resource Center, Bloomington, IN, USA, stock # 1089) for heat-shock inducible expression in *Drosophila* cells, LR-recombination reactions were performed with 75 ng of donor and 75 ng of acceptor in a 5 μL reaction with LR recombinase (Thermo Fisher Scientific, Waltham, MA, USA, Gateway LR Clonase II, #11791020). according to the manufacturer's protocol. Plasmids for transfection were prepared with a Qiagen maxiprep kit (Hilden, Germany).

Cell Culture: *Drosophila* Schneider 2 (S2) cells (Expression Systems, Davis, CA, USA #94-005F) were cultured in ESF media (ESF 921 Insect Cell Culture Medium, Expression Systems) with 5% fetal bovine serum (FBS, Wisent, St. Bruno, QC, Canada at room temperature on plates. Cells were passaged every 2 to 3 days. On the night before transfection, $1.5 \times 10^6$ S2 cells were plated per well of 6-well plates. The next day, the media were changed and a transfection mix was added. Mirus Transit insect Transfection Reagent (Mirus Bio, Madison, WI, USA) was used for the transfections according to the manufacturer's protocol.

To mark nuclei, pAct5C-H2Av-RFP (gift of V. Archambault) was co-transfected with Venus constructs. The day after transfection, the media were changed, and the next day, cells were replated on a Concanavalin A-coated glass-bottom imaging dish (Ibidi, Munich, Germany). Cells were heat-shocked for 8 min at 37 °C to induce Ph expression and used the next day for live imaging. Transfection efficiency was assessed from one experiment with Venus-Ph, for which phase contrast images were also collected. We carried out a manual analysis of these images, counting all visible cells, all RFP+ cells, and all Venus+ cells. This analysis indicated a 55 +/− 9% transfection efficiency based on H2Av-RFP (*n* = 2546 cells in 10 images). Only 33 +/− 10% of the RFP+ cells were also Venus+, or 18 +/− 6% of the total.

Western Blotting: 500,000 S2 cells per well were plated in a 24-well plate (Corning 353 047) one day prior to transfection. Fresh media were added the next day, and transfection was carried out as described above. The next day, the media were changed, and the evening of the following day, cells were heat-shocked for 8 min at 37 °C. After 24 h, transfected S2 cells were counted, and 500,000 cells were centrifuged at 1295× *g* at 4 °C for 5 min. Pellets were re-suspended in 70 μL of 2× SDS-PAGE buffer (232 μL/mL of Tris pH 6.8, 100 μL/mL of glycerol, 34 mg/mL of sodium dodecyl sulfate (SDS), and 120 mg/mL of bromophenol blue) and boiled for 5 min. Samples were then run on 8% and 16% SDS-PAGE gels for Ph and H2A-RFP, respectively, for 80 min at 120 volts before being transferred to nitrocellulose membranes.

Membranes were blocked for 30 min in 5% milk/PBST (1XPBS, 0.3% TWEEN® 20) and incubated overnight at 4 °C on a shaker in primary antibodies diluted in 5% milk/PBST. The primary antibodies used are as follows: anti-α-tubulin (mouse, 1:3000, Sigma Aldrich, Oakville, ON, Canada #T5168), anti-Ph (rabbit, 1:3000, Francis lab), anti-GFP (rabbit, 1:3000, Cedarlane labs, Burlington, ON, Canada #50430-2-AP), anti-RFP (rabbit, 1:3000, St. Johns Laboratory, London UK #STJ97083) and anti-H2B (mouse, 1:3000, Abcam, Toronto, ON, Canada, ab52484). Membranes were washed 3 times for 10 min each in PBST, incubated for 2 h in secondary antibodies diluted in 5% milk/PBST, and washed 3 times again for 10 min in PBST. Secondary antibodies were conjugated to Alexa Fluor 680 (anti-rabbit and anti-mouse, Thermo Fisher Scientific #A21076 and #A21057, respectively) or 800 CW (anti-Rabbit, Li-Cor, Lincoln, NE, USA) and used at 1:25,000 in 5% milk/PBST. Blots were scanned on an Odyssey CLx imager. The anti-Ph antiserum was raised against aa773-984 of Ph-p and was used previously described [26]. It was not affinity-purified. This antigen has been used by the Müller lab to generate anti-Ph antisera [70].

Cellular Fractionation: Cellular fractionation was carried out as in [46,47]. Transfections were carried out as described above in 6-well plates. Cells were allowed to grow for up to 6 days after transfection and then heat-shocked to induce expression, as described above, 16–20 h before harvesting. Between 1 and $1.7 \times 10^7$ cells were used for each fractionation. All centrifugations were carried out at 4 °C, and all procedures were carried out on ice. Cells were pelleted at 1300× *g* in a JS5.3 rotor for 4 min and washed with 1 mL of ice-cold PBS; 5% of cells were removed for total cell extracts, and the remaining cells were centrifuged as described above. Pellets were resuspended in 500 μL of Buffer A (10 mM of HEPES, pH 7.9, 10 mM of KCl, 1.5 mM of $MgCl_2$, 0.34 M of sucrose, 10% glycerol, 0.1% TritonX-100, and 1 mM of DTT) with protease and phosphatase inhibitors as indicated (0.2 mM of PMSF, 10 μg/mL of aprotinin, 10 μg/mL of leupeptin, 2 μg/mL of pepstatin, 16 μg/mL of benzamidine, 10 μg/mL of phenanthroline, 50 μg/mL of TLCK, 50 mM of NaF, 2 mM of Na-ortho-vanadate, and 40 mM of β-glycerophosphate). Cells were incubated on ice for 5 min and pelleted for 4 min at 1300× *g*. The supernatant was collected as S1; pellets were washed once in 500 μL of Buffer A and pelleted again. Pellets were resuspended in 500 μL of Buffer B (3 mM of EDTA, 0.2 mM of EGTA, and 1 mM of DTT), with the same phosphatase and protease inhibitors as indicated for Buffer A, and then they were incubated 30 min on ice. Nuclei were pelleted by centrifugation at 1700× *g*, and the supernatant was collected as S3. Pellets were resuspended in 350 μL of DNaseI digestion buffer (10 mM of Tris, pH 7.6, 2.5 mM of $MgCl_2$, and 0.5 mM of $CaCl_2$) with 20 μg of RNaseA and 2 μL of DNaseI per sample and incubated on ice for 1 h. Nuclei were

centrifuged at full speed in a refrigerated microfuge. The supernatant was saved as S4, and chromatin pellets were resuspended in 2X SDS-sample buffer as P2. An analysis of the histone distribution in fractions indicated that the nuclease digestion was not successful, since histones were not released into S4. We therefore pooled signals from S4 and P2 as "chromatin-associated". 6X-SDS-sample buffer was added to S1, S3, and S4 to 1X. All samples were boiled for at least 5 min and stored at −20. Western blot analysis was carried out as described above using 10% SDS-PAGE.

Image Acquisition: Live images were acquired with a Zeiss microscope equipped with a Yokogawa CSU-1 spinning-disk confocal head. A 63X oil objective was used, and the software for image acquisition was Zen 2012. The excitation wavelengths for Venus and RFP were 488 and 561 nm, respectively. For the VENUS channel, the laser power and exposure were set as 2.40% and 77 mS, respectively. For the RFP channel, they were 11% and 500 mS, respectively. For live imaging, $3 \times 3$ tiles of confocal stacks of 1 μm slices were collected.

Image Processing and Analysis (CellProfiler): Live images were opened using ImageJ (Fiji) in the tiff format. A maximum intensity projection was made for each image. Images were then split into VENUS and RFP channels, which were named the foci and nuclei, respectively. These were then uploaded to Cell Profiler (3.1.9). Cell Profiler modules were used to build the basic analysis pipeline, but a few parameters were modified for different constructs. The basic pipeline modules were the identification of nuclei from RFP staining and foci identification from the VENUS channel. The module IdentifyPrimaryObjects was used for the identification of the nucleus, which was named "nuclei". Objects outside a diameter range and touching the image border were excluded. Global thresholding was applied with either the Minimum Cross Entropy or Otsu algorithms. The identification of foci was conducted using the IdentifyPrimaryObjects module, with thresholding using Otsu or Minimum Cross Entropy, and named "foci". The size of the smoothing filter and the distance between local maxima parameters were separately adjusted for each construct to segment clumped objects. In the case of constructs such as PhΔ1 where extensive clumping was observed, EnhanceorSuppressFeatures was used to ignore highly clumped objects. The RelateObjectsModule was used to count the number of foci per nucleus. MeasureObject-Intensity was used to obtain mean intensity values. For the mean VENUS intensity, the nuclei objects corresponding to the nuclei images were selected. MeasureObjectSizeShape provided the measurements of size, which is the total number of pixels within the object. The object selected in this case was foci. For condensates that have complex morphologies, are small, or are faint relative to surrounding levels (such as those formed by PhΔ1), the software does not detect them well and often fuses multiple tiny condensates, so the numbers are likely underestimates.

For live image analysis, at least 100 cells were analyzed per experiment for each construct. All observations were made from at least three experiments. Statistics were calculated using GraphPad Prism v8.4.3, using recommended settings, including for correction for multiple comparisons. *p* values were calculated for Kruskal–Wallis tests with Dunn's multiple-comparison correction.

Manual Image analysis (ImageJ): To test the relationship between protein levels and condensate formation and to confirm the results from automated analysis, we analyzed between 48 and 480 cells using a single confocal slice in ImageJ (Fiji). We selected cells with visible signals in both the Venus and H2Av-RFP channels. Nuclei were segmented by hand using the magic wand tool. In less than 10% of cases, we instead used the freehand drawing tool if the magic wand could not capture the nucleus (typically if two cells were close together). The mean intensity in nuclei was then measured in the Venus channel. To count condensates, we used "find maxima", and assigned them to each measured nucleus as described here: https://microscopy.duke.edu/guides/count-nuclear-foci-ImageJ (accessed on 23 May 2022). The find maxima "prominence" parameter was adjusted for each construct—in general, constructs that formed small foci were well-captured with prominence set to 1000, and a prominence of 5000 or 10,000 was used for constructs that

formed large condensates (to prevent the identification of multiple maxima in single condensates). These data were used to plot the mean intensities for different constructs and to compare the intensity for cells with and without condensates. The same data were used to create our image galleries, where 8 cells from the same image were selected and ordered by mean intensity to illustrate the relationship between total protein levels and condensate formation. The images were adjusted so that both the H2A-RFP and Venus signals are visible in all cases, so the intensity cannot be compared based on the images shown. The 7th or 8th most intense cell was used to plot the profile of Venus and H2A-RFP intensity across at least one condensate (a line was drawn through the condensate and "plot profile" used in both channels) to qualitatively assess the colocalization of condensates with chromatin.

**Supplementary Materials:** The following supporting information can be downloaded at: https://www.mdpi.com/article/10.3390/epigenomes6040040/s1. Figure S1 Predicted disorder in Polyhomeotic-like Ph homologues; Figure S2 Complete sequences of Ph IDRs; Figure S3 Transfected proteins are (over)expressed as full-length proteins; Figure S4 Summary of effects of Ph IDRs on condensate formation; Figure S5 Full gels of Western blots of cell fractionation; Figure S6 Quantification of Ph IDR expression levels in transfected cells.

**Author Contributions:** Conceptualization, I.K. and N.J.F.; formal analysis, I.K. and N.J.F.; funding acquisition, N.J.F.; investigation, I.K., E.L.B. and N.J.F.; project administration, N.J.F.; supervision, N.J.F.; writing—original draft, I.K. and N.J.F.; writing—review and editing, N.J.F. All authors have read and agreed to the published version of the manuscript.

**Funding:** This work was funded by a grant from the Canadian Institutes for Health Research (CIHR) to N.J.F.

**Institutional Review Board Statement:** Not applicable.

**Informed Consent Statement:** Not applicable.

**Data Availability Statement:** The datasets generated during the current study are available from the corresponding author on reasonable request.

**Acknowledgments:** I.K. thanks members of the Francis lab, particularly Djamouna Sihou, for critical input on the project.

**Conflicts of Interest:** The authors declare no conflict of interest.

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
