# Peer review of "Regulation of Polyhomeotic Condensates by Intrinsically Disordered Sequences That Affect Chromatin Binding"

_2075-4655, 2020_

Round 1
Reviewer 1 Report
In this paper, Kapur et al. used transfection experiments with different Ph parts to understand the role of not-SAM regions of Ph that are required for proper condensate formation. This work is a continuation of previously published data from Francis’s lab (Seif et al., 2020, Nat Comm, doi: 10.1038/s41467-020-19435-z.). The authors have a large data set that is carefully presented and controlled. The data are of high quality, organized well to highlight several interesting observations. The text is well written and comprehensive, integrating the author's new findings with existing knowledge. I think this work will be an interesting and important resource contribution? to the field.
I can recommend this article for publication in Epigenomes.
However, I have several minor comments that should be answered prior to publication:
Abstract: Please, decipher the abbreviation ‘IDR’ in the abstract
70: in both Drosophila and mammalian cells, and in mice – it’s not clear what the difference between the ‘mammalian cells’ and ‘mice’
176: Average total Ph levels did not exceed 2-fold of untransfected cells, due to downregulation of endogenous ph. The Ph locus is known to contain Polycomb Response Elements [41], which may mediate auto-regulation [42, 43]. – what the percent of S2 cells were transfected? I suggest that it was not 100% and you really had the mixed population of transfected and not-transfected cells. If so, the level of downregulation of endogenous ph should be higher than 2-fold. Could you indicate the transfection efficiency in Materials and Methods section or, optimally, insert the Fig in Supplementary? Сould downregulation of endogenous ph be related to cell stress upon the transfection?
578: anti-Ph (rabbit, 1:3 000, Francis lab) – please, add here the reference if anti-Ph antibodies were used in previously published articles or add the description of antibodies
Discussion: I suggest that authors could also indicate that SAM domain are not unique for Ph. Moreover, at least two other proteins in Drosophila, Scm and Sfmbt, also contain C-terminal SAM domains and are involved in PcG-dependent silencing. Could SAM domains of Scm and Sfmbt be also implicated in phase separation/chromatin binding/3D nuclear organization? Please, briefly discuss it.
Author Response
REVIEWER 1
In this paper, Kapur et al. used transfection experiments with different Ph parts to understand the role of not-SAM regions of Ph that are required for proper condensate formation. This work is a continuation of previously published data from Francis’s lab (Seif et al., 2020, Nat Comm, doi: 10.1038/s41467-020-19435-z.). The authors have a large data set that is carefully presented and controlled. The data are of high quality, organized well to highlight several interesting observations. The text is well written and comprehensive, integrating the author's new findings with existing knowledge. I think this work will be an interesting and important resource contribution? to the field.
Abstract: Please, decipher the abbreviation ‘IDR’ in the abstract--done
70: in both Drosophila and mammalian cells, and in mice – it’s not clear what the difference between the ‘mammalian cells’ and ‘mice’—We have added “cultured cells” and “developing mice” to clarify the difference between cultured cells and a developing organism (73-74)
176: Average total Ph levels did not exceed 2-fold of untransfected cells, due to downregulation of endogenous ph. The Ph locus is known to contain Polycomb Response Elements [41], which may mediate auto-regulation [42, 43]. – what the percent of S2 cells were transfected? I suggest that it was not 100% and you really had the mixed population of transfected and not-transfected cells. If so, the level of downregulation of endogenous ph should be higher than 2-fold. Could you indicate the transfection efficiency in Materials and Methods section or, optimally, insert the Fig in Supplementary? Сould downregulation of endogenous ph be related to cell stress upon the transfection?
Unfortunately we do not have data on the transfection efficiency for most of the experiments. From the live imaging, we only have transfected H2Av-RFP, and Venus-Ph. We have one data set for Venus-Ph for which we also have phase contrast images. We carried out a manual analysis of these images, counting all visible cells, all RFP+ cells, and Venus+ cells. This analysis indicates a 55 +/-9% transfection efficiency based on H2Av-RFP (n=2546 cells in 10 images). Only 33+/-10% of the RFP+ cells were also Venus+, or 18+/-6% of the total. This may reflect lower activity of the HS promoter (our heat shock is very mild, 8 min. at 37 degrees). We have added these numbers to the methods (643-645).
We are not sure we can say much about the expected level of endogenous Ph if there is some down-regulation by the transfected proteins because of the wide range of Ph levels observed in different cells (>one order of magnitude in most cases). We have expanded the explanation in the text to qualify that we observe changes in Ph levels; they could be related to the autoregulation implied by the presence of PREs in the Ph regulatory region, but we cannot rule out other possibilities (like a response to transfection as the reviewer suggests)(190-194).
578: anti-Ph (rabbit, 1:3 000, Francis lab) – please, add here the reference if anti-Ph antibodies were used in previously published articles or add the description of antibodies
We have added these details to the methods (665-667).
Discussion: I suggest that authors could also indicate that SAM domain are not unique for Ph. Moreover, at least two other proteins in Drosophila, Scm and Sfmbt, also contain C-terminal SAM domains and are involved in PcG-dependent silencing. Could SAM domains of Scm and Sfmbt be also implicated in phase separation/chromatin binding/3D nuclear organization? Please, briefly discuss it.
-We have added this important point to the discussion (526-539)
Reviewer 2 Report
The authors demonstrated that Polyhomeotic condensates are regulated by their intrinsically disordered regions (IDR, located at the N-terminal part of the protein) that affect chromatin binding. Polyhomeotic is part of the Polycomb Group multi protein complexe PRC1, a well known repressive complex involved in the repression of many developmental genes via the histone marks H2A119Ub and H3K27me3. Here, they used the Drosophila Polyhomeotic which has homologs in mammals, named PHC. Because of the conservation of the Polycomb regulatory system, the regulation of condensate formation maybe functionally conserved.
Globally, the paper is well written and well presented. Their cell-based characterization of the Ph IDRs is rather exhaustive and the results often support their conclusions, with the caveat that this is an over-expression in cell system that will need to be further confirmed by in vitro studies.
Here, I bring few points that need to be addressed by the authors before publication.
Major points:
- They claim that PH1 is enriched in Serine (S), but contrary to PH3, we don’t really see it on Fig.2D. Is there an explanation for this? Histidine (H) and Glutamine (Q) might also be important in PH1.
- Cytoplasmic condensates are observed in the Venus-WT Ph, for instance in nuclei 1-4 of Fig. 2B. This is an observation obvious on their image, but this is actually very intriguing considering the repressive function of Polycomb proteins in the nucleus. Is it something relevant for protein function? In other word, do they see this with their anti-Ph in untransfected cells or alternatively, is it just trivially explained by protein over-expression?
- To measure the threshold for condensate formation, they use the mean intensity (Fig.3G, 4G and 5G). It is however difficult to relate these graphs with their single cell analysis (Fig.3A,C, 4A,C and 5A,B). In other words, are nuclei 1 to 8 pairwised comparable in term of intensity? For example, it could be nice to have the quantified Venus intensity for each represented 1 to 8 individual nucleus. If so, all acquisitions must be performed with the same settings and quantification cannot be done on saturated images/pixels.
Alternatively, the authors could present a new panel of 2 nuclei per conditions that have the same Venus intensity in order to appreciate the differences in condensate formation between the different constructions.
- They insist on the fact that Ph IDRs may play their role by affecting interaction with chromatin. To support this, would it be possible to perform some ChIP experiments at known Polycomb target loci with relevant constructs, using the Venus and YFP Tags for the pull-down. This could also help to define if some of the observed condensates could be functional or not.
Minor points:
- Fig.1D: The blue colors of PH1 and PH3 look very (too) similar.
- Fig.3G, 4G and 5G, it would be nice to have the comparison with the WT PH directly included in the graph, as in the case of panel E and F of the same figures.
- There is a typo in the labelling of Fig.5A and 5B: Top, replace Venus-PhΔ2 by Venus-PhΔ3 and Venus-PhΔ1Δ3 by Venus-PhΔ1Δ2, respectively.
- Again in Figure 5, follow the same logic as in Fig.3 and 4; i.e. 5A, 5C>5B, 5B>5C, 5D…
- In Fig.7, could have been nice to have the percentage of cells that form condensates.
- I realized with the n mentioned in the legend of Fig.7 that this percentage is in general very low, so maybe change the following sentence:
line 364: « …, condensates form without the SAM although the number of cells that form condensates is low » to « …, condensates form without the SAM although the number of cells that form condensates is very low »
- In Fig.8, same remark as for Fig.7, could be nice to have the percentage of cells that form condensates.
- I found the Table 1 more confusing than helpful. Actually, it could help to distinguish small and large condensates and have a sort of quantification, i.e. for example + versus +++.
Author Response
REVIEWER 2
The authors demonstrated that Polyhomeotic condensates are regulated by their intrinsically disordered regions (IDR, located at the N-terminal part of the protein) that affect chromatin binding. Polyhomeotic is part of the Polycomb Group multi protein complexe PRC1, a well known repressive complex involved in the repression of many developmental genes via the histone marks H2A119Ub and H3K27me3. Here, they used the Drosophila Polyhomeotic which has homologs in mammals, named PHC. Because of the conservation of the Polycomb regulatory system, the regulation of condensate formation maybe functionally conserved.
Globally, the paper is well written and well presented. Their cell-based characterization of the Ph IDRs is rather exhaustive and the results often support their conclusions, with the caveat that this is an over-expression in cell system that will need to be further confirmed by in vitro studies.
Here, I bring few points that need to be addressed by the authors before publication.
Major points:
- They claim that PH1 is enriched in Serine (S), but contrary to PH3, we don’t really see it on Fig.2D. Is there an explanation for this? Histidine (H) and Glutamine (Q) might also be important in PH1.
We have clarified that S is not over-represented, but it is masked by the CAST algorithm, which indicates that it is repetitive (contributing to low complexity). We also note the slight enrichment for Q and H, and agree with the reviewer that these are notable features. (143-146).
- Cytoplasmic condensates are observed in the Venus-WT Ph, for instance in nuclei 1-4 of Fig. 2B. This is an observation obvious on their image, but this is actually very intriguing considering the repressive function of Polycomb proteins in the nucleus. Is it something relevant for protein function? In other word, do they see this with their anti-Ph in untransfected cells or alternatively, is it just trivially explained by protein over-expression?
We do not see clear cytoplasmic condensates with anti-Ph staining, and they have not been reported for Drosophila expressing GFP-Ph (which is mild overexpression) (Netter et al., 2001). We noted it in the text because while cytoplasmic condensates are certainly triggered by overexpression, they are nevertheless construct specific—we observe them consistently with WT-Ph and with PhD3, and only occasionally with other constructs, despite the fact that overexpression levels are similar. Our results indicate that Ph can form condensates in the cytoplasm (i.e. chromatin binding is not required). We think there are 2 possible mechanisms—either Ph normally shuttles between nucleus and cytoplasm (which is, as the reviewer points out, potentially intriguing for function), or Ph condensates that form in the cytoplasm may not be able to enter the nucleus. These possibilities need to be investigated in detail, including at physiological expression levels. We agree with the reviewer though that this could hint at a cytoplasmic function for Ph, or a role for nuclear import in regulating condensates. We have added text to the discussion (540-550) to flag these observations as potentially worthy of follow up.
- To measure the threshold for condensate formation, they use the mean intensity (Fig.3G, 4G and 5G). It is however difficult to relate these graphs with their single cell analysis (Fig.3A,C, 4A,C and 5A,B). In other words, are nuclei 1 to 8 pairwised comparable in term of intensity? For example, it could be nice to have the quantified Venus intensity for each represented 1 to 8 individual nucleus. If so, all acquisitions must be performed with the same settings and quantification cannot be done on saturated images/pixels.
Alternatively, the authors could present a new panel of 2 nuclei per conditions that have the same Venus intensity in order to appreciate the differences in condensate formation between the different constructions.
The cells shown in each figure are from images taken with identical settings. They are not directly comparable across constructs as they were chosen to span the range of intensities for the cell population analyzed in the single image from which the cells were chosen. To contextualize the images with the quantification, we have indicated the 8 cells shown in the figure on the graphs of mean intensity (orange dots) (Fig. 2F, 3G, 4G, 5G).
- They insist on the fact that Ph IDRs may play their role by affecting interaction with chromatin. To support this, would it be possible to perform some ChIP experiments at known Polycomb target loci with relevant constructs, using the Venus and YFP Tags for the pull-down. This could also help to define if some of the observed condensates could be functional or not.
This is a good experiment, and an important one, but it is beyond the scope of the work presented, which is already quite extensive. We also prefer to pursue ChIP experiments with more homogeneous and physiological expression levels—the wide range of expression levels we see would make if very difficult to relate ChIP observations to condensates observed by microscopy. We have noted this in the Discussion (577-584).
Minor points:
- Fig.1D: The blue colors of PH1 and PH3 look very (too) similar.
We agree with the reviewer and have changed the colour of Ph3 in all of the figures.
- Fig.3G, 4G and 5G, it would be nice to have the comparison with the WT PH directly included in the graph, as in the case of panel E and F of the same figures.
We are not able to do this because the images used for WT Ph and mini-Ph were taken with different settings (they are part of our previously published analysis). The images used for the intensity analysis (3G, 4G, and 5G) were collected with the same settings; we have compiled all the intensity results for these images into one graph for comparison, which is (new) Figure S4G.
- There is a typo in the labelling of Fig.5A and 5B: Top, replace Venus-PhΔ2 by Venus-PhΔ3 and Venus-PhΔ1Δ3 by Venus-PhΔ1Δ2, respectively.
- Again in Figure 5, follow the same logic as in Fig.3 and 4; i.e. 5A, 5C>5B, 5B>5C, 5D…
We apologize for these errors and have corrected them.
- In Fig.7, could have been nice to have the percentage of cells that form condensates.
- I realized with the n mentioned in the legend of Fig.7 that this percentage is in general very low, so maybe change the following sentence:
line 364: « …, condensates form without the SAM although the number of cells that form condensates is low » to « …, condensates form without the SAM although the number of cells that form condensates is very low »
We have added “very low” to the text as suggested, We have compiled the fraction of cells that form condensates for each analysis, which is presented in (new) sFig. 4F for the IDR deletions, and Fig. 7C and 8C for the ΔSAM and IDRs alone.
- In Fig.8, same remark as for Fig.7, could be nice to have the percentage of cells that form condensates. Please see above response.
- I found the Table 1 more confusing than helpful. Actually, it could help to distinguish small and large condensates and have a sort of quantification, i.e. for example + versus +++.
We have added a “size” column to the table, and a more quantitative assessment of condensates as suggested.